# Pregnane X receptor agonist nomilin extends lifespan and healthspan in preclinical models through detoxification functions

Shengjie Fan [1,8], Yingxuan Yan[1,8], Ying Xia[2,3,8], Zhenyu Zhou[1,8], Lingling Luo[1,8], Mengnan Zhu[4,5,8], Yongli Han[1], Deqiang Yao[6], Lijun Zhang[1], Minglv Fang[1], Lina Peng[4,5], Jing Yu[1], Ying Liu[1], Xiaoyan Gao[1], Huida Guan[7], Hongli Li[1], Changhong Wang [7], Xiaojun Wu[7], Huanhu Zhu[4]✉, Yu Cao [2,3]✉ & Cheng Huang [1]✉

Citrus fruit has long been considered a healthy food, but its role and detailed mechanism in lifespan extension are not clear. Here, by using the nematode *C. elegans*, we identified that nomilin, a bitter-taste limoloid that is enriched in citrus, significantly extended the animals' lifespan, healthspan, and toxin resistance. Further analyses indicate that this ageing inhibiting activity depended on the insulin-like pathway DAF-2/DAF-16 and nuclear hormone receptors NHR-8/DAF-12. Moreover, the human pregnane X receptor (hPXR) was identified as the mammalian counterpart of NHR-8/DAF-12 and X-ray crystallography showed that nomilin directly binds with hPXR. The hPXR mutations that prevented nomilin binding blocked the activity of nomilin both in mammalian cells and in *C. elegans*. Finally, dietary nomilin supplementation improved healthspan and lifespan in D-galactose- and doxorubicin-induced senescent mice as well as in male senescence accelerated mice prone 8 (SAMP8) mice, and induced a longevity gene signature similar to that of most longevity interventions in the liver of bile-duct-ligation male mice. Taken together, we identified that nomilin may extend lifespan and healthspan in animals via the activation of PXR mediated detoxification functions.

Delaying the ageing process is one of the major aims of modern biomedical research. Manipulation of multiple signalling pathways or dietary restriction has been shown to extend the lifespan and healthspan in animal models, but these methods are either not practical or not satisfactory for application to the general population[1–6]. Increasing evidence has shown that the expression of detoxification enzyme genes and resistance to toxins are increased in long-lived flies, worms and rodents[7]. Detoxification gene expression is increased in the liver and shows more resistance to hepatotoxins in long-lived Little mice, Ames dwarf mice, Snell dwarf mice, growth hormone receptor

[1]School of Pharmacy, Shanghai University of Traditional Chinese Medicine, Shanghai 201203, China. [2]Department of Orthopaedics, Shanghai Key Laboratory of Orthopaedic Implant, Shanghai Ninth People's Hospital, Shanghai Jiao Tong University School of Medicine, Shanghai 200011, China. [3]Institute of Precision Medicine, the Ninth People's Hospital, Shanghai Jiao Tong University School of Medicine, 115 Jinzun Road, Shanghai 200125, China. [4]School of Life Science and Technology, ShanghaiTech University, Shanghai 201210, China. [5]CAS Center for Excellence in Molecular Cell Science; Shanghai Institute of Biochemistry and Cell Biology, Chinese Academy of Sciences, Shanghai; University of Chinese Academy of Sciences, Beijing 100049, China. [6]iHuman Institute, ShanghaiTech University, Shanghai 201210, China. [7]Institute of Chinese Materia Medica, Shanghai University of Traditional Chinese Medicine, Shanghai 201203, China. [8]These authors contributed equally: Shengjie Fan, Yingxuan Yan, Ying Xia, Zhenyu Zhou, Lingling Luo, Mengnan Zhu. ✉e-mail: zhuhh1@shanghaitech.edu.cn; yu.cao@shsmu.edu.cn; chuang@shutcm.edu.cn

knockout mice[8–11], and dietary and methionine-restriction mice[12–15]. This phenomenon is also observed in long-lived *Caenorhabditis. elegans (C. elegans)* and *Drosophila melanogaster*, which are also more resistant to xenobiotics[16–18]. Recently, Tyshkovskiy et al. have shown that the expression of drug metabolism and detoxification genes, such as *cytochrome P450 enzymes* (*CYPs*) and *glutathione-S-transferases* (*GSTs*), was increased in the livers of mice that underwent 17 known lifespan-extending interventions, indicating that targeting detoxification may be a useful longevity intervention therapy[19].

The expression of xenobiotic detoxification enzyme genes is transcriptionally regulated by nuclear hormone receptors (NHR). In mammals, pregnane X receptor (PXR) is a major regulator of the expression of drug metabolism and xenobiotic detoxification genes[20]. In *C. elegans*, NHR8 and DAF-12 transcriptionally regulate the expression of these genes in order to excrete toxins. NHR-8 and DAF-12 are required for the longer lifespan and healthspan in *C. elegans*[1,21–24], indicating that NHR-8 and DAF-12 are required for the longevity of *C. elegans*. Hence, we propose that the activation of the NHRs mediating detoxification gene expression may be a strategy for lifespan-extending intervention and ageing-related diseases.

Nomilin is a naturally-occurring compound in *citrus* fruits such as lemons, grapefruits, oranges as well as in tangerine seed and peel[25,26]. A number of studies have showed that nomilin may exert a variety of pharmacological properties including anti-cancer, anti-inflammatory, anti-obesity, anti-viral, anti-oxidant, immune-modulatory and neuroprotective effects[25,26]. Here, we show that nomilin is a PXR agonist, and may extend lifespan and healthspan in *C. elegans* and mice via NHR-regulated detoxification functions, and induce the common transcriptome markers seen in the liver of mice in response to most lifespan-extending interventions.

## Results

### Nomilin extends lifespan in wild type (WT) *C. elegans*
Recently, certain metabolites/small molecules have been shown to have a potentially useful ageing inhibiting ability in the nematode *C. elegans*, and have been reported to have a similar effect in mammals[27–30]. Inspired by those findings, we searched for components present in oranges that have a longevity intervention effect, since orange extracts have been reported to extend lifespan and healthspan in *C. elegans*[31,32]. Among many known components, we were particularly interested in nomilin, a limonoid enriched in citrus fruits[26], because it has also been suggested to have certain health-promoting and disease-preventing properties[33–37]. Surprisingly, we found that nomilin extended the lifespan of WT N2 *C. elegans* in a dose-dependent manner. Treatment with 25, 50 and 100 μM nomilin significantly increased the average lifespan by 9.4%, 24% and 24%, respectively (Fig. 1a, & Supplementary Table 1). However, when the concentration was increased to 200 μM, nomilin showed the lower lifespan extending effects, implying that higher concentration of nomilin may have a side effect on *C. elegans* (Supplementary Fig. S1a, Supplementary Table S2). Then, we compared the lifespan-extending effects of nomilin and its analogue limonin. Limonin displayed less effects on the survival time when compared to nomilin (Supplementary Fig. S1b, Supplementary Table S2), indicating the structure specificity of nomilin. In addition, the accumulation of lipofuscin, a biomarker of senescence in *C. elegans*, was also significantly reduced under nomilin supplementation (Supplementary Fig. S1c, d). Locomotion behaviours in aged adults (which have been commonly used to analyse the ageing-related healthspan of *C. elegans*), such as body-bend, head-swing, and pharynx-pumping, were also significantly improved under nomilin treatment (Supplementary Fig. S1e-h). Moreover, like many long-lived *C. elegans* models, nomilin-treated animals also showed increased resistance to heat and oxidative stress (Supplementary Fig. S1i, j). Taken together, these data suggest that nomilin may delay the ageing process and extend lifespan and health span in *C. elegans*.

It has been reported that reduction of food intake or dead bacteria feeding may extend lifespan in *C. elegans*[38,39]. To exclude the possibility that the lifespan-extending effects of nomilin were due to reduced food uptake, we performed three experiments. First, we found that nomilin supplementation did not affect the growth speed of *E. coli* OP50 (Supplementary Fig. S1k), the lab food of *C. elegans*. Second, we assessed the lifespan of animals grown on heat-killed OP50, and found that dead bacteria extended the lifespan of worms and nomilin further increased the lifespan (Supplementary Fig. S1l, Supplementary Table S2). Third, a food-taxing experiment showed that *C. elegans* did not avoid the nomilin-supplemented bacteria lawn (Supplementary Fig. S1m, n). These data suggest that the lifespan-extending effects of nomilin are not likely to result from the reduction of food-intake or suppression of bacteria growth. Moreover, because infertility may extend the worm lifespan, we checked the average brood size and the offspring number of worms and found that there were no significant differences between nomilin-treated and control animals (Supplementary Fig. S1o, p). Thus, these results indicate that nomilin extends *C. elegans* lifespan and healthspan directly.

### DAF-2 and DAF-16 are required for the extending lifespan effect of nomilin
We then tested which specific signalling pathway plays a major role in nomilin-associated lifespan extension. The insulin/insulin-like growth factor signalling (IIS) pathway plays essential roles in longevity and the resistance of the body to various stressors, such as oxidative stress and xenobiotic stress, in *C. elegans*[40,41]. We found that nomilin could not further extend the lifespan in *C. elegans* insulin-like peptide receptor mutant *daf-2(-)* (Fig. 1b). Moreover, nomilin supplementation significantly promoted nuclear translocation of DAF-16::GFP, a FOXO transcription factor downstream of DAF-2 IIS[42] (Fig. 1c, d), and *daf-16(-)* also completely blocked the lifespan extension effect of nomilin (Fig. 1e). To confirm that DAF-16 signalling is involved in nomilin function, the mRNA levels of DAF-16 downstream targets were tested. The results showed that the mRNA levels of *sod-2*, *sod-3*, *clk-2* and *lin-2* were increased by nomilin (Fig. 1f). Mutation of *C. elegans* histone deacetylase SIRT/*sir-2.1* (Fig. 1f) or mTORC1 signalling component RagA/B homologue *raga-1* (Fig. 1g), another two well-known ageing pathways, could not fully block the lifespan extension effect of nomilin. Interestingly, we found that nomilin did not enhance the dauer formation either in the WT or in the *daf-2(e1370)* mutant background (Supplementary Fig. S1q, r). We thought that the reason normilin mainly affected longevity instead of dauer formation, possibly because it targeted the intestinal cells and affected the local IIS activity (Fig. 1c, d). It was consistent with the report that the intestinal IIS pathway mainly regulates longevity, but not the dauer formation process, while the neuronal IIS pathway does the opposite[43]. These data suggest that the lifespan-extending effects of nomilin in *C. elegans* mainly depend on the intestinal IIS pathway.

### Nomilin activates detoxification enzymes and protects *C. elegans from* multiple toxins
To better understand the mechanistic role of nomilin in the lifespan extension process, we performed a literature search and found several studies that suggest that nomilin can enhance the expression of the detoxification genes glutathione S-transferase and quinone reductase that, which are responsible for the detoxification of quinone-containing compounds in the liver and intestine of rodents[44,45]. To investigate whether these detoxification roles of nomilin are conserved in *C. elegans*, we examined the detoxification gene expression under nomilin supplementation in worms using real-time reverse transcription-polymerase chain reaction (RT-PCR). As expected, the expression of multiple detoxification genes was increased, such as phase I genes *cyp35a3-5* and *cyp37a1*; phase II genes *gst-4* and *ugt44*; and phase III genes *gpg-3* and *gpg-12/13* (Fig. 2a). We further tested whether nomilin could protect *C. elegans* from toxins such as

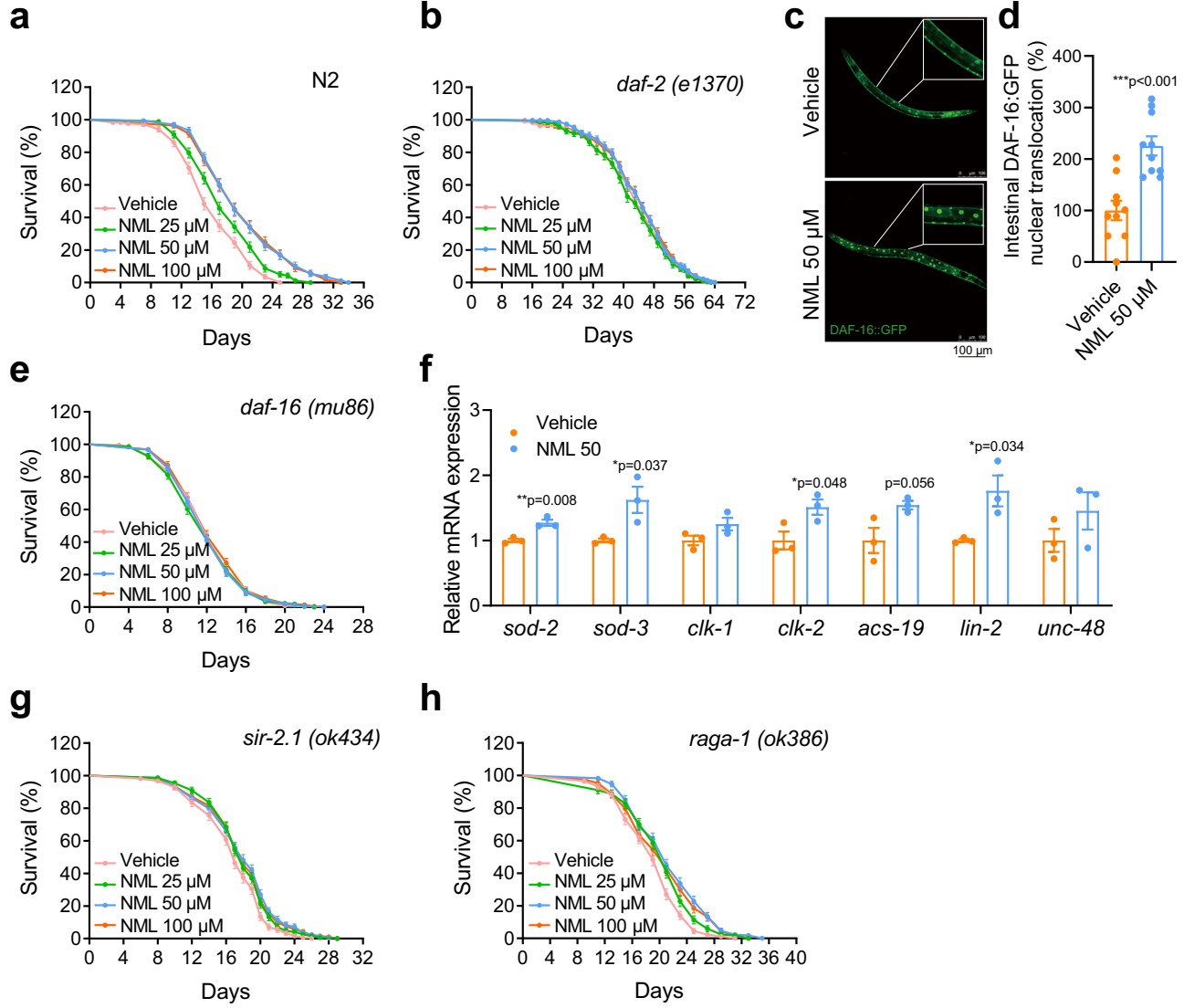

**Fig. 1 | Nomilin extends the lifespan of *C. elegans* via a *daf-2/daf-16*-dependent pathway. a, b** Lifespan curves showing the lifespan-extending effects of nomilin at various concentrations on WT N2 *C. elegans* (**a**), *daf-2(e1370)* (**b**). Nomilin significantly extended the lifespan of WT, but not the *daf-2*, *daf-16*, and *sir-2.1* animals. *raga-1* partially blocked the lifespan extension effect of nomilin. **c** Representative fluorescent pictures showing the DAF-16::GFP nuclear localisation in control (upper) and nomilin-treated (bottom) animals. **d** The statistical data of (**c**, two-

tailed unpaired Student's *t*-test, *n* = 10/each, ***p* < 0.001 vs. control group). **e** Lifespan curves of nomilin in *daf-16 (mu86)*. **f** Quantification of mRNA levels of *daf-16* downstream genes in *C. elegans* (two-tailed unpaired Student's *t*-test, *n* = 3/each). **g, h** Lifespan curves of nomilin in *sir-2.1(ok434)* (**g**), and *raga-1(ok386)* mutants (**h**). All data were expressed as mean ± SEM. Detailed information is shown in Supplementary Table S1.

colchicine, chloroquine, paraquat and methyl mercury chloride (MeHgCl), which are involved in the ageing process and the pathogenesis of senile diseases[46–50]. Animals survived better under nomilin treatment in a dose-dependent manner (Fig. 2b–e), indicating that, in addition to lifespan extension, nomilin also protects the worms from many toxins.

### Nomilin extends lifespan and improves toxin resistance via nuclear hormone receptors NHR-8 and DAF-12

We then tried to identify the direct target of nomilin in *C. elegans* through which it exerts its lifespan extension and detoxification abilities. Although the InR/DAF-2 pathway is known to increase lifespan and toxin resistance[51,52], it is highly unlikely that DAF-2 binds with nomilin directly, given its nature as an insulin-like peptide receptor[41]. Instead, nomilin may function via binding with certain nuclear hormone receptors (named NHR hereafter), a large family of proteins that can interact with small metabolites and regulate metabolism and other

physiological functions. From the literature review, we found that two NHRs (NHR-8 and DAF-12) have been reported to play major roles in both lifespan extension and detoxification[53,54]. We then tested the role of these two NHRs during nomilin treatment. Surprisingly, compared to N2 worms (Fig. 2f), we found that both the loss-of-function mutation of the *daf-12* worm (*rh61rh411*) (which causes the loss of the capacity for ligand binding and DNA binding)[55] and *nhr-8* (*tm1800*) fully suppressed the lifespan extension effect of nomilin (Fig. 2g–h, Supplementary Table S3). In addition, we found the detoxification effect of nomilin also depended either on NHR-8 or DAF-12 (or both, in the case of methyl mercury chloride) (Fig. 2b–e). To confirm that the action of nomilin was mediated by *nhr-8* and *daf-12*, we crossed the *muls109* males with *daf-12* (*rh61rh411*) and *nhr-8* (*tm1800*) hermaphrodites and obtained a homozygous strain of Pdaf-16::gfp::*nhr8* and Pdaf-16::gfp::*daf-12* worms, who were then treated with nomilin. The results showed that nomilin did not promote the nuclear translocation of DAF-16 in *daf-12* and *nhr-8* mutant worms (Fig. 2i–j).

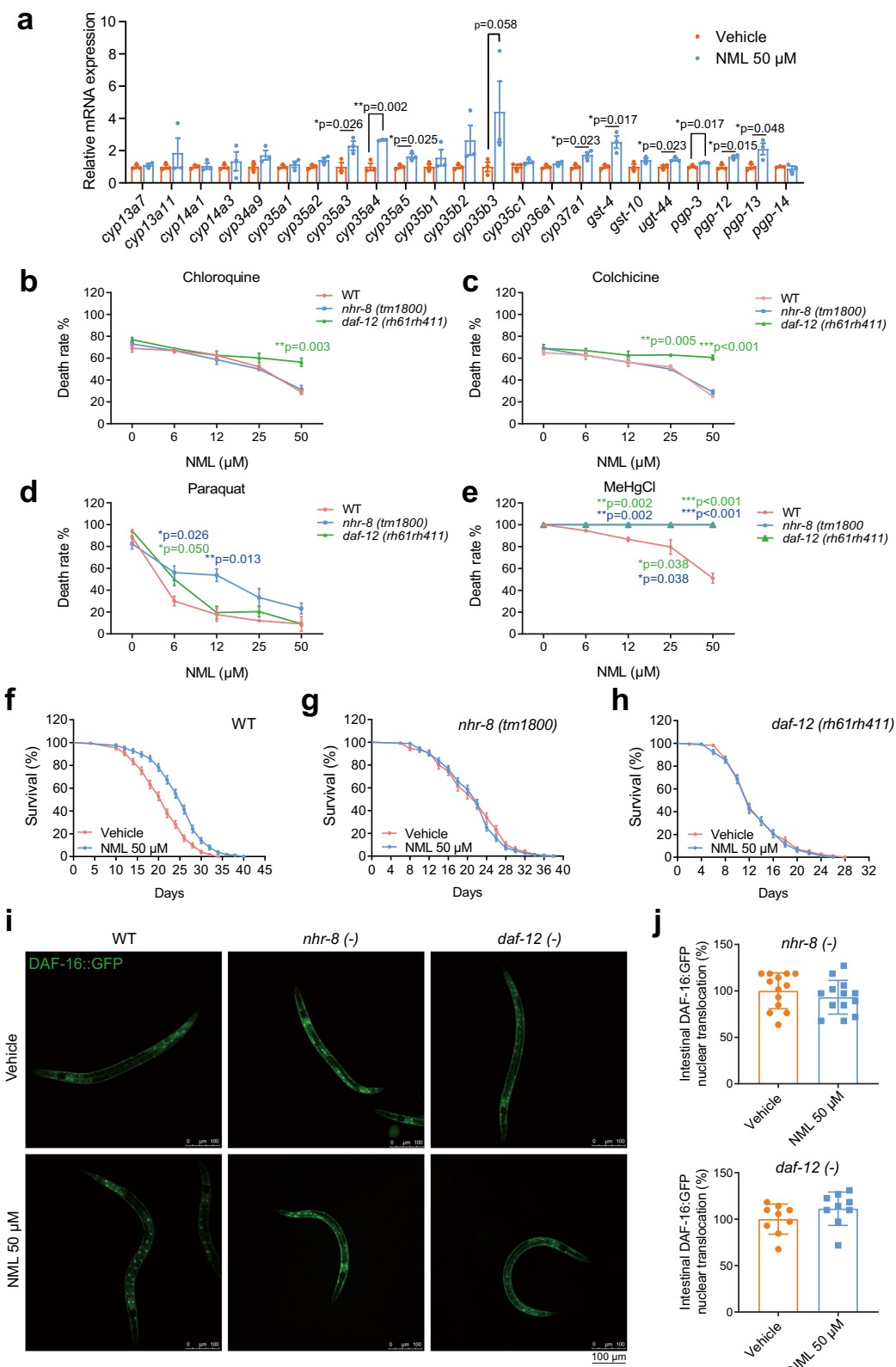

Next, we investigated whether the downstream detoxification enzymes of *daf-12* and *nhr-8* are involved in the lifespan-extending effects of nomilin. Indeed, upregulation of most detoxification genes by nomilin (Fig. 2a) was blocked in *nhr-8* and *daf-12* mutant (Fig. 3a, b), indicating that these genes are the targets of *nhr-8* and *daf-12*. Then, nomilin-activated genes *gst-4*, *cyp35a3*, *pgp-3* and *pgp-14* were knocked down using RNAi in N2 worms, who were then treated with nomilin.

The results showed that the lifespan-extending effects of nomilin were attenuated in *gst-4*, *cyp35a3* and *pgp-3* knockdown worms when compared to those under RNAi treatment, while deficiency of *pgp-14* that was not regulated by nomilin (Fig. 2a) did not change the effects of nomilin (Fig. 3c–g, Supplementary Table S4). Collectively, these data indicate that nomilin extends lifespan through the targeting of nuclear hormone receptors NHR-8 and DAF-12.

**Fig. 2 | Nomilin executes its ageing inhibiting and detoxification abilities via nuclear hormone receptors *nhr-8/daf-12* in *C. elegans*. a** Quantification of mRNA levels of detoxifying genes in *C. elegans* (two-tailed unpaired Student's *t*-test, *n* = 3/each, each sample contains about 1000 worms). **b**–**e** Survival curves showing the protective effects of nomilin on worms with the indicated genotypes upon various chemical toxin treatments (two-way ANOVA test, *n* = 3/each, ***p* < 0.001). Nomilin-treated *C. elegans* were more resistant to chloroquine (**b**), colchicine (**c**), paraquat (**d**), and MeHgCl (**e**) than wild type N2, but not *nhr-8* (*tm1800*) or *daf-12* (*rh61rh411*)

animals. **f**–**h** lifespan curves showing the lifespan-extending effects of nomilin on WT, *nhr-8* mutant (*tm1800*), and *daf-12* mutant (*rh61rh411*) *C. elegans*. The detailed information is shown in Supplementary Table S3. **i** Effects of nomilin on nuclear trans-localisation of *nhr8::daf-16::GFP* and *daf12::daf-16::GFP* worms. The worms were treated with 50 μM of nomilin from L1 to L4, and 10 animals were examined per condition. **j** Average number of cells with *DAF-16::GFP* nuclear localisation in *nhr-8* and *daf-12* mutants. All data were expressed as mean ± SEM, ***p* < 0.001 vs. control group, *n* = 9 or 13 worms per group.

## IIS signalling is involved in detoxification in *C. elegans*

The upregulation of xenobiotic detoxification genes is a common characteristic of long-lived flies, worms and rodents, some of which showed stronger resistance to xenobiotic stressors. It has been proposed that the resistance to a broad range of stressors, including heat, oxidative stress and xenobiotics, could be a longevity-assurance mechanism. In *C. elegans*, several lines of evidence have suggested that detoxification and longevity are coupled. Long-lived *daf-2* mutants showed a similar transcriptomic signature of increased detoxification gene expression to flies and mice; however, resistance to toxins in *daf-2* mutants has not been studied to date. Thus, we explored the detoxification functions of *daf-2* and *daf-16* mutants. Under challenge with paraquat or MeHgCl, *daf-2* mutants were more resistant than WT worms. In contrast, short-lived *daf-16* mutants were more sensitive (Supplementary Fig. S2a, b), suggesting that IIS signalling may be involved in xenobiotic detoxification.

To investigate the correlation between *daf-2* and *nhr-8/daf-12* in detoxification, we carried out RNAi against *nhr-8* or *daf-12* in *daf-2* worms (*daf-2::nhr-8* RNAi, *daf-2::daf-12* RNAi), and then challenged with paraquat or MeHgCl. The results showed that deficiency of *nhr-8* and *daf-12* diminished the detoxification ability of *daf-2* worms (Supplementary Fig. S2a, b), indicating that the detoxification ability of *nhr-8/daf-12* and IIS longevity signalling may show crosstalk with the lifespan extension function.

## Nomilin is a specific PXR agonist

Next, we tried to identify whether there is a mammalian counterpart of *C. elegans* NHR-8/DAF-12. Previous reports have suggested that NHR-8/DAF-12 belong to the NR1 subfamily, a group of NHRs specifically functioning in xenobiotic metabolism[7]. Because of the limited sequence homology between mammalian and *C. elegans* NHRs, we applied an HEK293 cell-based reporter assay to test which mammalian NHRs could be activated by nomilin. Among nine well-known human NR1 subfamily NHRs (hPPARα, hPPARβ, hPPARγ, FXR, LXRα/β, NRF2, hCAR and hPXR), only hPXR could be effectively activated by nomilin (Fig. 4a, Supplementary Fig. S3a–i). The activation effect of nomilin was similar to the known strong hPXR agonist rifampicin (Rif). Moreover, the nomilin analogue deacetylnomilin (which lacks an acetyl group at the hydroxyl residue on $^2$C) displayed similar activation effects (or moderately stronger), while another analogue, limonin (which lacks the iconic heptatomic lactone ring), did not have any activity (Fig. 4a), suggesting that the heptatomic lactone ring of nomilin may be essential for binding to hPXR. In addition, the Time-resolved fluorescence resonance energy transfer (TR-FRET) assay showed that the binding between the labelled hPXR ligand and hPXR-LBD was significantly attenuated by nomilin in a dose-dependent manner (IC50: 5.8 μM and Kd: 13.3 μM), more so than by potent PXR agonist T0901317 (IC50: 11.7 nM and Kd: 30.1 μM), while deacetylnomilin showed lower affinity (IC50: 22.7 μM and Kd: 198.3 μM, Fig. 4b), suggesting that nomilin may compete with the labelled ligand to bind with hPXR. Together, these data suggest that nomilin is a specific agonistic ligand of hPXR, the potential mammalian ortholog of NHR-8/DAF-12.

## The crystal structure of the nomilin-hPXR complex indicates critical amino acids for the binding affinity

To further confirm the direct binding between nomilin and hPXR, and discover more information on its structure and activation mechanism, we expressed the ligand-binding domian (LBD) domain of hPXR (residues 130–432) fused with residues 676–700 of nuclear receptor coactivator 1 (NCOA1$^{676-700}$) at the C-terminal as a co-activator peptide. The purified hPXR$^{LBD}$-NCOA1$^{676-700}$ (hPXR chimera hereafter) was co-crystallised with nomilin and the structure of the protein-drug complex was determined at a resolution of 2.1 Å, allowing a detailed observation of the drug-target interaction (Fig. 4c and Supplementary Fig. 3j, k). The hPXR chimera forms a homodimer in the purification and crystal structure, as previously reported. The dimeric hPXR forms an interface via the β1' strand of the characteristic antiparallel β-sheet[56–58]. A region of non-proteinaceous electron density was identified in the Fo-Fc map of the LBDs of each of the hPXR protomers, and the molecular framework of nomilin fit the electron density very well (Fig. 4d). The electron density in protomer A was refined with better continuity than that in protomer B (Fig. 4e). As shown in Fig. 4f, g, the binding of nomilin in the hPXR chimera was majorly mediated by the hydrogen bonds formed by two carbonyl oxygens with S247 and Q285, as well as the strong hydrophobic interaction among the backbone carbons of nomilin, and the cavity formed by a series of hydrophobic residues including M243, W299, I414 and M425 (Fig. 4g and Supplementary Fig. 3l, m). To exclude that nomilin was modified during the crystallization process, the hPXR/nomilin crystal was assayed by mass spectrometry. The result did not show different structure in the crystal.

Two methionine residues, M243 and M425, closely contact both nomilin and rifampicin in their crystal structures, respectively[59]. In a functional assay of hPXR mutants, the binding of nomilin is abolished by the mutation of M425 and the function of rifampicin relies more on the M243 residue. From Fig. 4h, the biphenyl moiety in rifampicin interacts with M243 more closely than nomilin does, making rifampicin more sensitive to the local spatial variation introduced by the M243Q mutation (Supplementary Table S5). In addition, one of the structural differences between hPXR bound with nomilin and rifampicin is the helix formed by amino acids 193–209, which is well-refined in our structure and previously reported structures of hPXR in complex with SR12813, clotrimazole, and hyperforin, but absent in the structure of hPXR-rifampicin complex (Supplementary Fig. S4a)[58,60–62]. The structural superposition between hPXR bound with nomilin and rifampicin showed the spatial clash between the biphenyl moiety of rifampicin and the helix$^{193–209}$ (Supplementary Fig. S4b), suggesting a displacement of the helix in the rifampicin-bound state. These data showed the LBD conformational change between nomilin-activated and rifampicin-activated states, which results in different biological effects.

To confirm the working model of hPXR, conservative and non-conservative mutations were made of the residues critical for nomilin binding, and a reporter gene assay was performed on these hPXR mutants. The transactivities of all mutant hPXR were lower than that of wild type hPXR, which may reflect the lower response of hPXR mutants to endogenous PXR agonists (Fig. 4i). M243A/Q did not affect the nomilin activity. However, S247A/R, M425A/Q and W299R completely blocked the activity of nomilin (Fig. 4i), indicating that these amino acid residues are critical for nomilin-dependent hPXR activation.

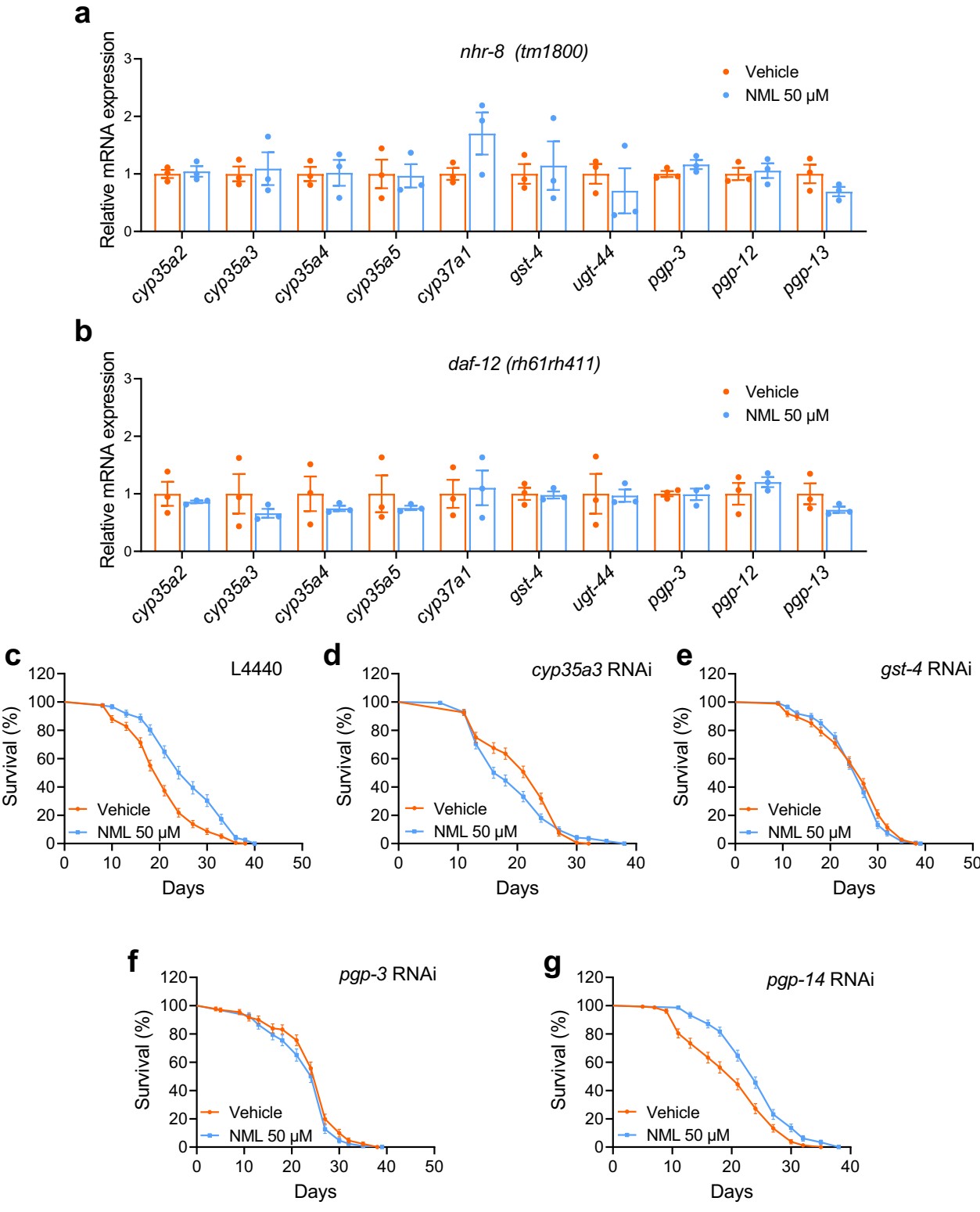

**Fig. 3 | Effects of nomilin on the detoxifying gene expression in *nhr-8* and *daf-12* mutant *C. elegans*. a, b** Quantification of mRNA level of genes in *nhr-8* (**a**) and *daf-12* (**b**) mutant *C. elegans*. (two-tailed unpaired Student's *t*-test, *n* = 3/each, each sample contains about 1000 worms). **c–g** Survival curve of nomilin in xenobiotic metabolism gene RNAi N2 *C. elegans*. Synchronised L1 worms were fed with *E. coli* (HT115) containing an empty control vector (L4440) until L4, then transferred to plates containing *cyp35a3*, *gst-4*, *pgp-3* or *pgp-14* RNAi HT115 with nomilin (50 μM) and DMSO (0.1%) as controls. All data were expressed as mean ± SEM. The detailed information is shown in Supplementary Table S4.

## Mammalian PXR is a functional ortholog of NHR-8 /DAF-12

Next, we attempted to verify whether mammalian PXR is a functional ortholog of NHR/DAF-12 that mediates lifespan extension and detoxification by nomilin. As expected, *nhr-8 and daf-12* mutation shortened lifespan in both nomilin-treated and control animals (Fig. 5a–c, Supplementary Table S6). We found that overexpression of WT *hPXR* could partially restore the lifespan effect of nomilin in *nhr-8* or *daf-12* mutant animals (Fig. 5a–c, Supplementary Table S6), while *hPXR*^S247R

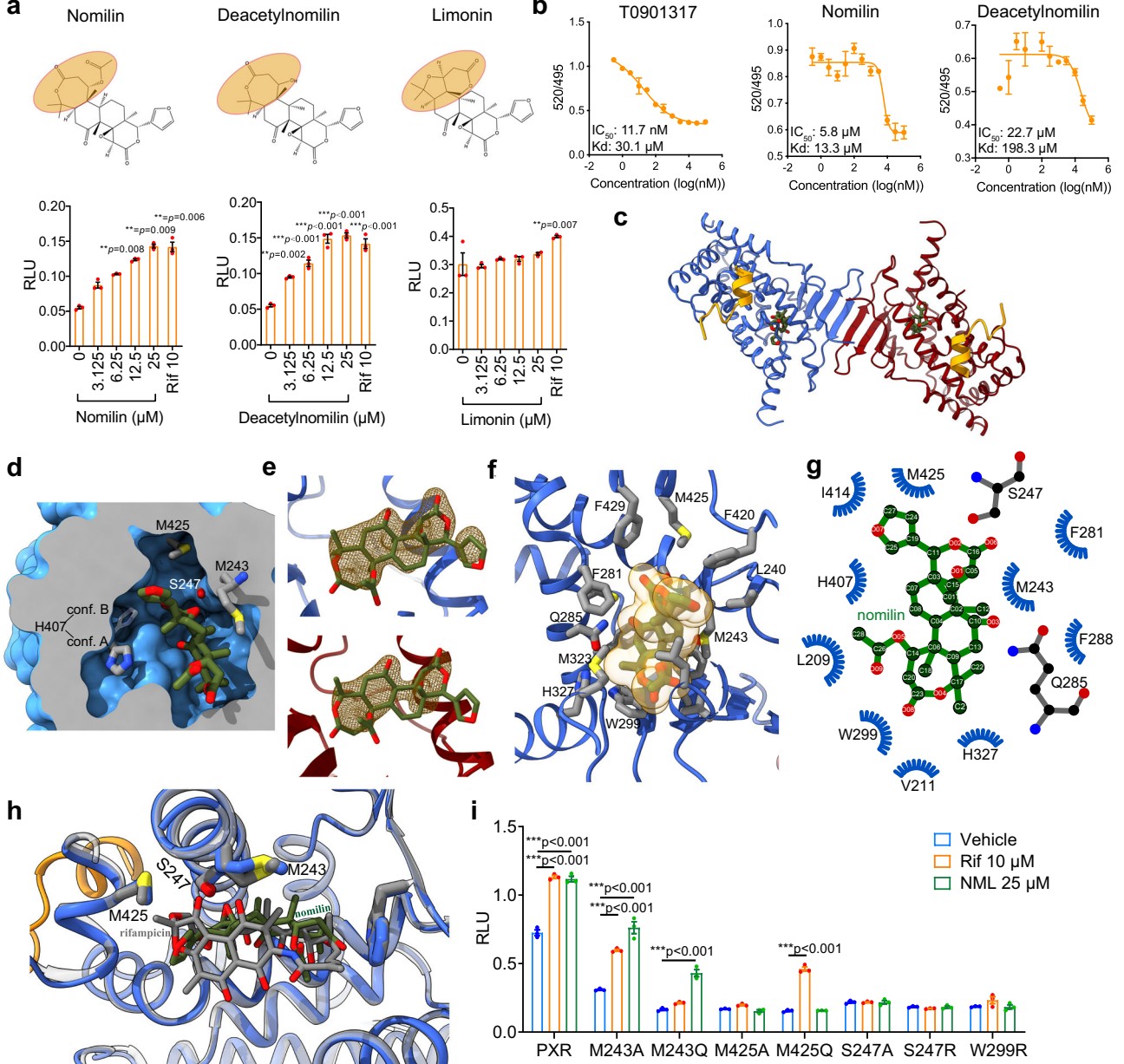

**Fig. 4 | Nomilin is a PXR agonist and the crystal structure for hPXR$^{LBD}$-NCOA1$^{676-700}$ bound to nomilin. a** Structure of nomilin and its analogues and hPXR reporter gene assay. The results represent three independent experiments. (One-way ANOVA test in deacetylnomilin and limonin; and Kruskal-Wallis test in nomilin, $n = 3$/each, the data were shown as means ± SEM. ***$p < 0.001$ compared to the control group). **b** TR-FRET assay. The TR-FRET ratio (520/495) was calculated by subtracting the background. **c** Dimeric human PXRLBD-NCOA1676-700 (in blue and red for protomer A and B, respectively) and co-activator peptide (in orange) fusion protein. The nomilin molecules are shown as stick models and coloured by element. **d** The space-filling model of protomer A was sliced to show the binding pocket for nomilin (shown as stick model) and some residues closely interacting with nomilin. H407 was shown in its two alternative conformations. **e** The omit Fo-Fc electron density maps for nomilin in protomer A (upper) and B (lower) are shown as mesh models and contoured to 1.0 σ. **f** A close view of the binding site of protomer A. The nomilin is shown as stick and space-filling models, with the surrounding residues shown as a stick model. **g** The schematic diagram for the hPXR-nomilin interaction network. **h** A comparison between binding pockets of hPXR LBD in complex with nomilin and rifampicin. Both hPXRs are shown as cartoon models in blue and grey for nomilin-bound and rifampicin-bound structures, respectively. The nomilin and rifampicin are shown as stick models coloured by element (green-red for nomilin and grey-red for rifampicin). The structure model of the hPXR-rifampicin complex was generated with coordinates from PDB ID 1SKX. **i** *hPXR* mutations change the effects of nomilin action. The plasmids were transfected into HEK293T cells, which were treated with nomilin or rifampicin for 24 h (two-tailed unpaired Student's *t*-test, $n = 3$/each, the data were shown as means ± SEM, ***$p < 0.001$ *vs.* control group).

(which blocks the binding between nomilin and hPXR) mutation only slightly restored the lifespan extension in *nhr-8*, and completely failed to restore the lifespan extension in *daf-12* mutants under nomilin treatment (Fig. 5a–c, Supplementary Table S6). The partial effect of hPXR to restore the lifespan extension effect of nomilin was possibly due to that mammalian PXR could not fully activate the *C. elegans* target genes. Moreover, to investigate whether hPXR could activate the target genes of NHR-8 and DAF-12, nomilin-treated *hPXR* transgenic *nhr-8* and *daf-12* worms were used to test mRNA levels. The results showed that nomilin only increased *gst-4*, *pgp-3* and *pgp-13* mRNA levels in *hPXR* transgenic *nhr-8* worms, and *pgp-13* mRNA in *hPXR* transgenic *daf-12* worms (Fig. 5d, e). These data indicate that hPXR could partially compensate for the function of NHR-8 and DAF-12 in mediating nomilin-dependent lifespan-extending effects in *C.*

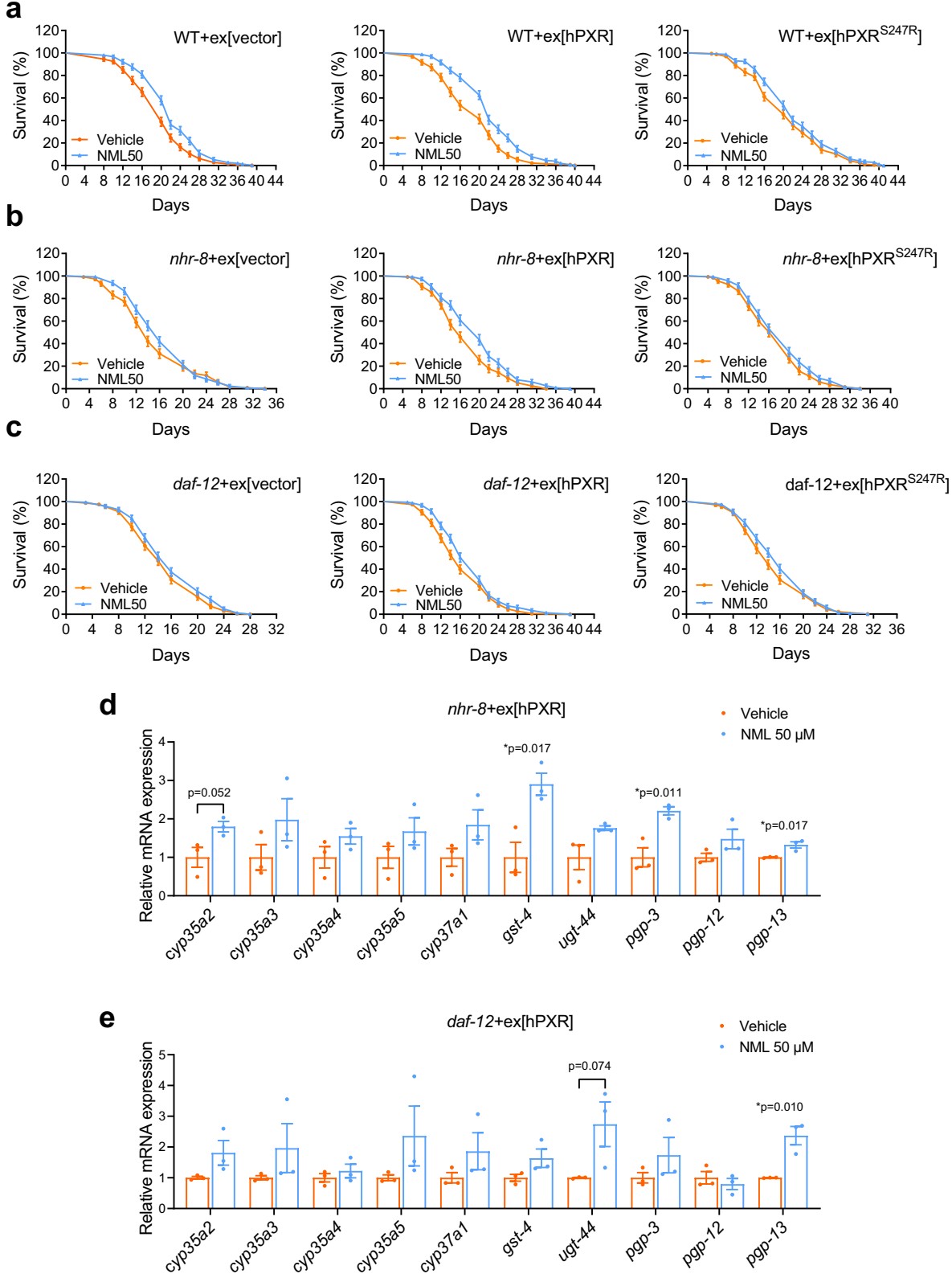

**Fig. 5 | *hPXR* partially restores the lifespan-extending effects of nomilin in *nhr-8* or *daf-12* mutants. a–c** lifespan curves showing the lifespan-extending effects of nomilin in *C. elegans* with indicated genotypes. Nomilin extended the lifespan in WT animals overexpressed with vector control (**a** left) or WT hPXR (**a** middle), but not hPXR^S247R (a hPXR mutant that blocked its binding with nomilin) (**a** right). Similarly, overexpression of WT hPXR (**b, c** middle), but not the vector control

(**b**, **c** left) or hPXR^S247R (**b, c** right), enabled the maximal lifespan extension of nomilin in *nhr-8* or *daf-12* animals. The detailed data are shown in Supplementary Table S6. **d, e**, The quantification of gene expression levels in *hPXR* transgenic *nhr-8* (**d**) and *daf-12* (**e**) mutant *C. elegans*. (Two-tailed unpaired Student's *t*-test, *n* = 3/ each, each sample contains about 1000 worms. The data were shown as means ± SEM).

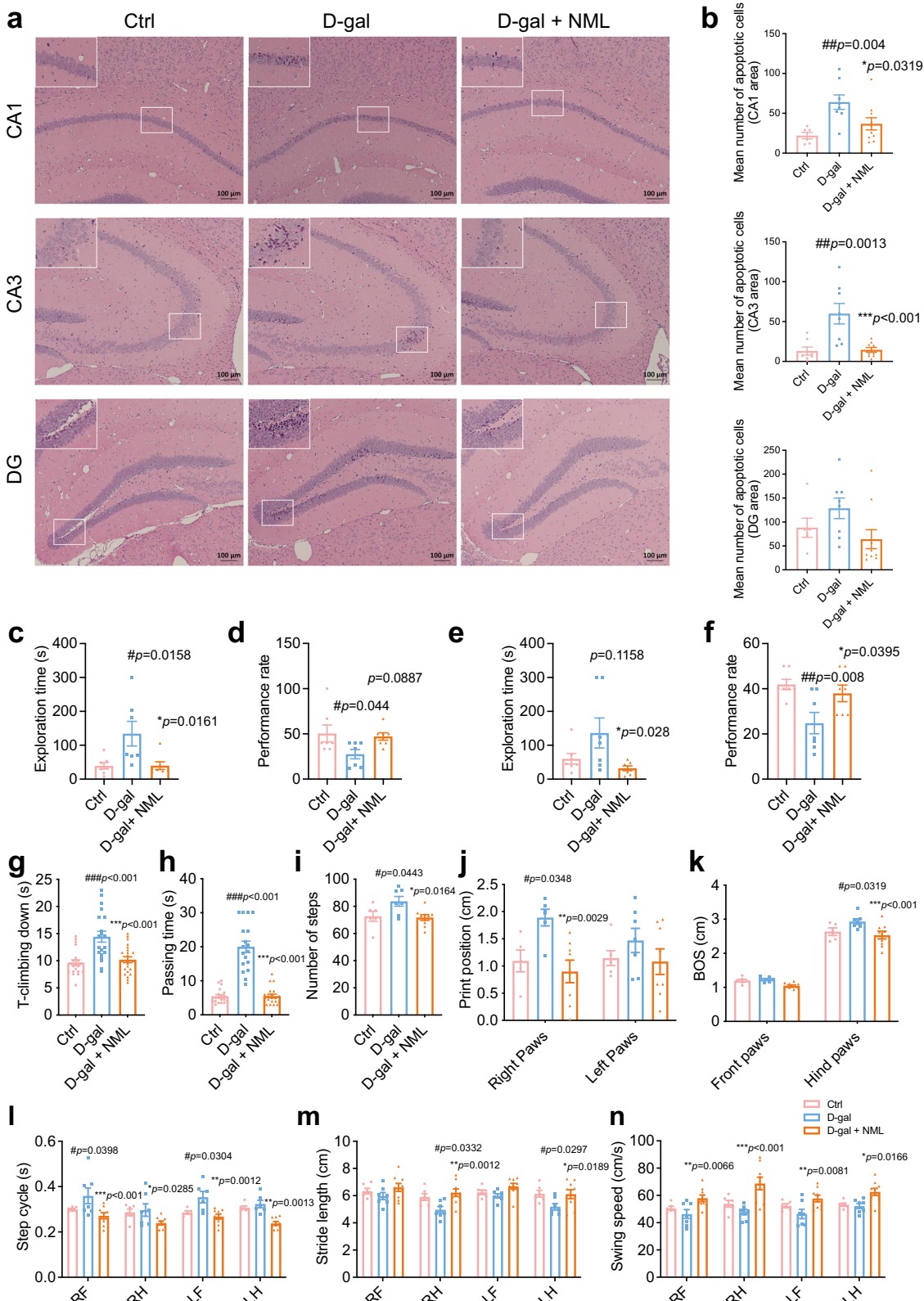

*elegans,* and the activation is dependent on the binding activity between nomilin and hPXR. It has been reported that PXR has multiple consensus DNA binding sequences including a relatively conserved 3' half-site A-G-T-T-C-A sequence[63]. Similarly, DAF-12 was also reported to have a similar 3' half-site A-G-T-T/G-C-A/G DNA binding sequence[64], and NHR-8 and DAF-12 share significant homology in DNA- and ligand-

binding domains (DBD; LBD), and have identical residues in the P-box, a motif in the first zinc finger that functions in DNA recognition[65]. Therefore, these may explain why hPXR could partially rescue the phenotypes of *daf-12(-)* mutant. Taken together, these data suggest that hPXR is an ortholog of NHR-8/DAF-12 and implicate nomilin in lifespan extension in mammals via the activation of hPXR.

**Fig. 6 | Effects of nomilin on apoptosis of the hippocampus, cognitive capacity and neuromuscular functions in D-galactose-induced mice. a** apoptotic cells in CA1, CA3 and the dentate gyrus of the hippocampus. **b** The quantitation of apoptotic cells in (**a**). ($n = 6$–7 for Ctrl, $n = 7$–8 for D-gal, $n = 9$–10 for D-gal+NML) Exploration time (**c**, $n = 7$/group) and performance rate (**d**, $n = 7$/group) in short-term memory test. Exploration time (**e**, $n = 7$/group) and performance rate (**f**, $n = 7$/group) in long-term memory test. **g** T-climbing down in pole test ($n = 20$ for Ctrl and D-gal, $n = 23$ for D-gal+NML). **h** Passing time in beam balance test ($n = 19$ for Ctrl, $n = 18$ for D-gal, $n = 21$ for D-gal+NML). The number of steps (**i**, $n = 6$ for Ctrl, $n = 7$ for D-gal, $n = 9$ for D-gal+NML), the print position (**j**, $n = 5$–6 for Ctrl, $n = 6$–7 for D-gal,

$n = 8$ for D-gal+NML), base of support (**k**, $n = 5$-6 for Ctrl, $n = 7$ for D-gal, $n = 9$ for D-gal+NML), step cycles (**l**, $n = 5$–6 for Ctrl, $n = 6$–7 for D-gal, $n = 9$ for D-gal+NML), stride length (**m**, $n = 6$ for Ctrl, $n = 7$ for D-gal, $n = 9$ for D-gal+NML) and swing speed (**n**, $n = 6$ for Ctrl, $n = 7$ for D-gal, $n = 9$ for D-gal+NML) in gait analysis. The mice were treated with D-galactose (125 mg/kg/day) and nomilin for 7 weeks. Scale bar = 100 μm (**a**). The data were shown as mean ± SEM. *p*-values were determined by one-way ANOVA test (**b–i**). *p*-values were determined by two-way ANOVA test (**j–n**). ###$p < 0.001$ *vs* the control group; ***$p < 0.001$ *vs* the D-galactose group. BOS base of support, NML nomilin, LF left forelimb, RF right forelimb, LH left hindlimb, RH right hindlimb, Ctrl Control, D-gal D-galactose, NML nomilin.

## Nomilin improves healthspan of D-galactose induced early-senescence mice

To test whether nomilin could improve healthspan in toxin-induced senescence, we used D-galactose to mimic the symptoms of human ageing in mice. D-galactose can be oxidized into hydrogen peroxide, which increases reactive oxygen species in cells, resulting in ageing of multiple organs[66,67]. D-galactose induced liver inflammation and the inflammatory cells infiltrated into the liver tissues (Supplementary Fig. S5a, b). The expression of inflammatory genes *Tnfα*, *Il-β* and *Mcp-1* was induced, and anti-oxidation genes *Ho-1*, *Nrf2* and *Sod-1* were suppressed in the liver of mice treated with D-galactose (Supplementary Fig. S5c, d). In contrast, nomilin significantly reversed these changes (Supplementary Fig. S5c, d).

Age-related damage was also observed in the central nervous system. D-galactose increased apoptotic cells in the CA1, CA3 and dentate gyrus in the hippocampus of the mice, which may result in neurodegeneration, while nomilin treatment reduced the numbers of dead cells (Fig. 6a, b). Cognitive decline is correlated with the change of the hippocampus during ageing. Thus, we adopted 8-arm maze to assess cognitive functions of the mice. In both short-term and long-term memory tests, the mean exploration time of D-galactose-treated mice was increased, while the performance rate was significantly reduced compared to those of control mice. However, the mean exploration time and the performance rate were reversed by nomilin treatment (Fig. 6c, d, e, f). The lower mobility resulting from impaired balance, lower stability and extremity strength is an age-related change in elders reflecting the functional decline of organs. The motor slowing in aged people is also commonly related to the structural and functional alterations of the elder brain[68]. Next, we performed the pole test and beam balance test in D-galactose-induced early-senescence mice; the T-climbing time in the pole test, and passing time in the beam balance test were longer than those in control mice (Fig. 6g, h). Following nomilin treatment, the times of T-climbing and passing time were reduced, equivalent to those of control mice (Fig. 6g, h), suggesting that nomilin could improve motor deficits in toxin-induced senescence mice. Gait analysis is a sensitive method for evaluating motor functions. Next, we used the Catwalk gait analysis system to assess whether nomilin could improve D-galactose-induced movement disorders. The number of steps, print position of both right and left paws, base of support (BOS) in the hind paws, and step cycles of the fore limbs were increased in D-galactose-induced mice, while stride length in the hind limbs and swing speed in the fore limbs of D-galactose mice were reduced, which were all reversed by nomilin treatment (Fig. 6i–n).

Then, we investigated whether nomilin activated mPXR downstream signalling in D-galactose-treated mice. The expression levels of mPXR downstream targets *Cyp3a11/13*, *Cyp2d22*, *Cyp2e1*, *Cyp8b1*, *Cyp51*, *Gsta1/2*, *Tyw1* and *Por* in the liver were upregulated by nomilin (Supplementary Fig. S6a), suggesting that it also activates mPXR in mice. To confirm that the mPXR target gene expression was increased by nomilin treatment, a Western blot was performed to assay the protein levels of Cyp3a11, Cyp51a1 and Gsta1. The results showed that the protein levels of Cyp3a11 and Gsta1 in the liver were increased by nomilin treatment (Supplementary Fig. S6b, c), supporting that mPXR

signalling was activated by nomilin. Taken together, the data suggest that nomilin may improve toxin-induced senescence, probably via the activation of detoxification function in mice.

PXR and its downstream detoxifying enzymes are also expressed in the brain[69–71]. Thus, we were curious to know whether nomilin can activate mPXR signalling in the hippocampus of D-galactose-treated mice. The analysis of mPXR target gene expression showed that the mRNA levels of *Gsta1*, *Mdr3*, *Cyp8b1* and *Cyp27a1* were reduced in the hippocampus, suggesting that the detoxification function was inhibited by D-galactose-treatment (Supplementary Fig. S6d). Nomilin significantly increased the mRNA levels of *Gsta1*, *Gsta2*, *Mdr3*, *Cyp8b1*, *Cyp27a1* and *Cyp2d22* in the hippocampus (Supplementary Fig. S6d). These data suggest that nomilin may also increase mPXR signalling in the brain of mice.

## PXR deficiency diminishes healthspan-extending effects of nomilin in mice

To confirm nomilin targeting of PXR, we analysed the healthspan-extending effects of nomilin in PXR knockout mice treated with D-galactose. The PXR[-/-] mice exhibited longer passing time in the beam balance test (Fig. 7a) and T-climbing time in the pole test (Fig. 7b). The dwelling time on the rotating rod test was reduced (Fig. 7c), while the number of falls from the rotating rod was increased (Fig. 7d) due to D-galactose treatment, which are similar to that in wild type mice. In gait test, the hind limb stance width was increased, and the hind limb stride length was decreased in D-gal treated PXR[-/-] mice, whereas the intervention of nomilin did not ameliorate gait instability in PXR[-/-] mice (Fig. 7e, f). The proportion of number of entries and exploration time of D-gal-induced mice were decreased compared to control mice, indicating D-gal-induced PXR[-/-] mice also have significant memory dysfunction, which is consistent with those in WT mice. However, memory dysfunction was not improved in the mice by nomilin treatment (Fig. 7g, h). D-galactose also resulted in neuron death in CA1 and CA3 of the hippocampus (Fig. 7i–l), and inflammatory infiltration in the liver of PXR[-/-] mice (Fig. 7m, n). However, nomilin did not improve the T-climbing time, the passing time, the dwelling time and the number of falls, or reduce cell death in the hippocampus and inflammatory cell infiltration in the liver of D-galactose-treated PXR[-/-] mice (Fig. 7a–n). These data suggest that nomilin could not improve the impaired motor mobility, neuron death and hepatic inflammation in D-galactose-induced senescent mPXR deficient mice, which further demonstrates that the healthspan-extending effects of nomilin occur via mPXR activation.

## Nomilin counteracts doxorubicin-induced senescence in mice

The chemotherapeutic drug doxorubicin may induce accelerated ageing and other long-term health conditions in cancer survivors[72–74]. This drug has been used to induce cellular and organ senescent in animal models[73,75]. Therefore, we assayed whether nomilin could extend the lifespan and healthspan in doxorubicin-treated mice. In the lifespan assay, the mice were treated with both doxorubicin and nomilin. Strikingly, the mean lifespan of nomilin-treated mice was extended by 50.57% (Fig. 8a, Fig. S7). In the healthspan experiments, doxorubicin increased the T-climbing time in the pole test and passing

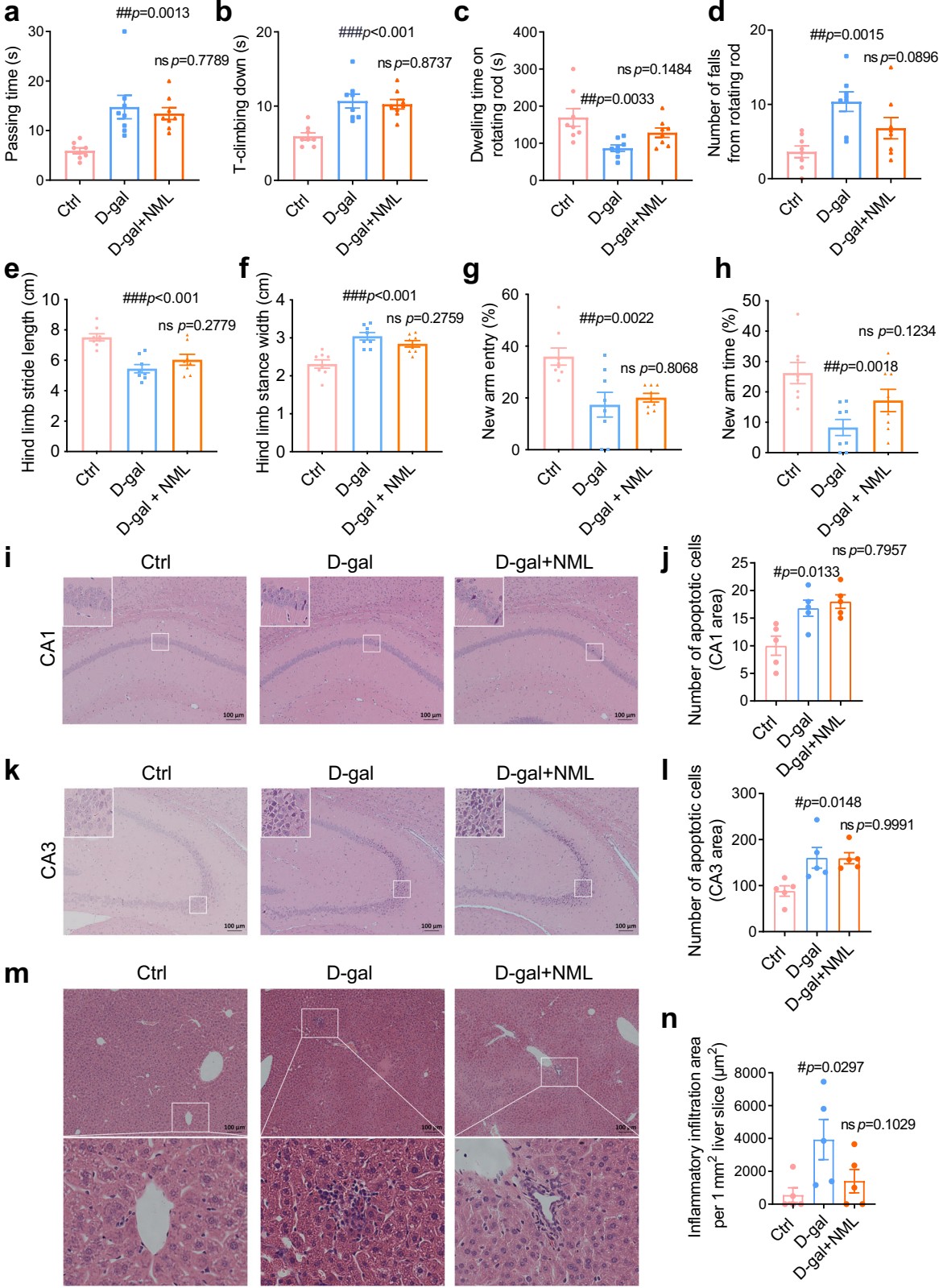

**Fig. 7 | Effects of nomilin on healthspan in D-galactose-treated PXR knockout mice. a** Passing time in beam balance test ($n = 8$/group). **b** T-climbing down in pole test ($n = 8$/group). **c** Dwelling time on rotating rod test ($n = 8$/group). **d** Number of falls from the rotating rod ($n = 8$/group). Hind limb stride length (**e**, $n = 8$/group) and hind limb stance width (**f**, $n = 8$/group) in gait analysis. **g** Proportion of the number of times entering new arm in the Y maze ($n = 8$/group). **h** Proportion of time spent in exploring new arm in Y maze ($n = 8$/group). **i, j** Apoptotic cells and the quantification of apoptotic cells in CA1 of the hippocampus ($n = 5$/group). **k, l** Apoptotic cells and the quantification of apoptotic cells in CA3 of the hippocampus ($n = 5$/group). **m** H&E staining of liver sections. **n** Inflammatory infiltration area per mm² liver sections ($n = 5$/group). The mice were treated with D-galactose (125 mg/kg/day) and nomilin for 7 weeks. The data were shown as mean ± SEM. *p*-values were determined by one-way ANOVA test (**a–h, j, l, m**). ###$p < 0.001$ *vs* the control group; ***$p < 0.001$ *vs* the D-galactose group.

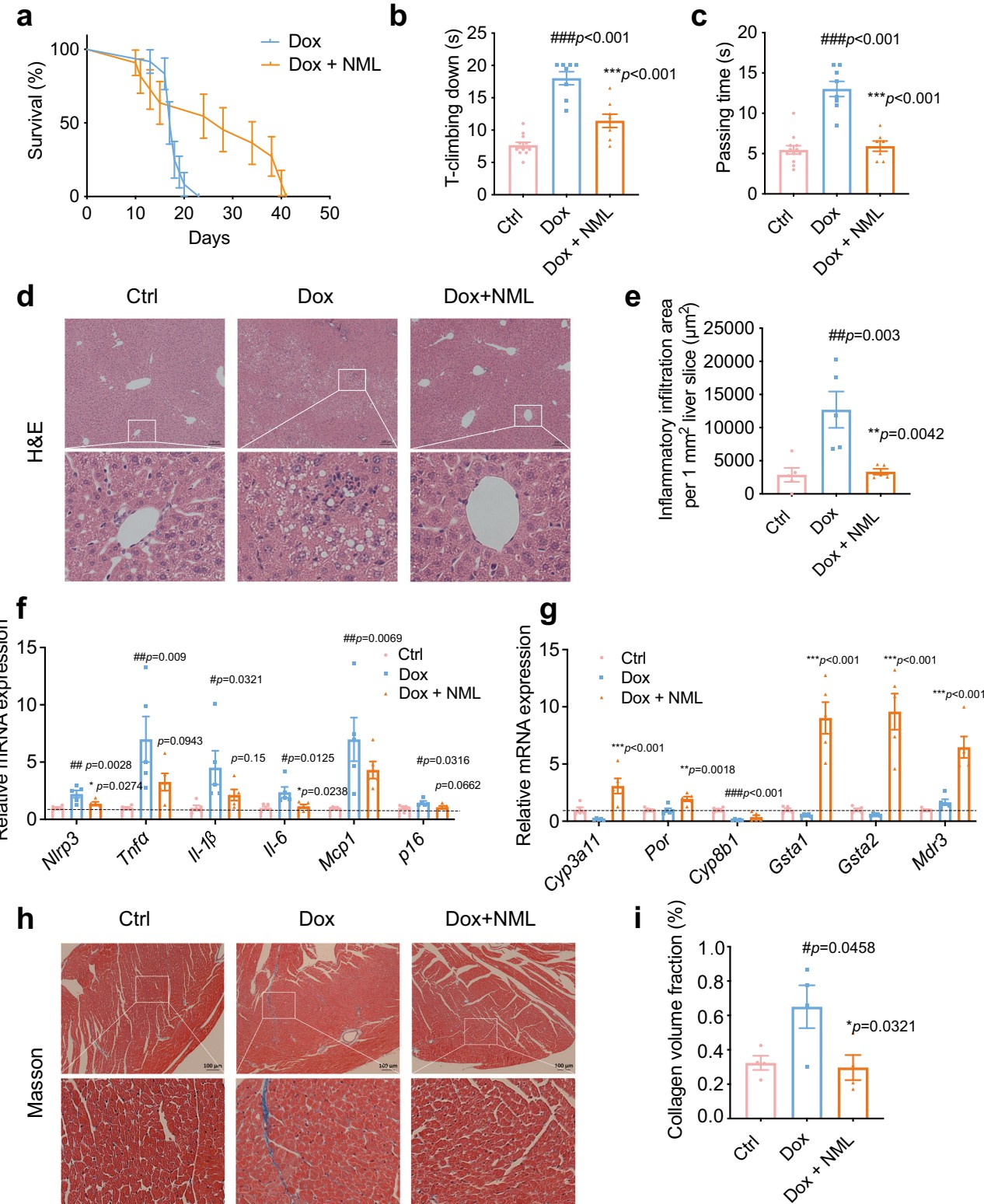

**Fig. 8 | Effects of nomilin on lifespan and healthspan in doxorubicin-induced senescence mice. a** Survival curve of accelerated ageing mice. The mice were treated with doxorubicin (5 mg/kg, three times a week) and nomilin (50 mg/kg/day). ($n = 12$ for Dox, $n = 11$ for Dox+NML) **b** T-climbing down in pole test ($n = 12$ for Ctrl, $n = 8$ for Dox and Dox+NML). **c** Passing time in beam balance test ($n = 12$ for Ctrl, $n = 8$ for Dox and Dox+NML). **d** H&E staining of the liver sections. **e** Inflammatory infiltration area per mm² liver sections in (**d**), ($n = 5$/group). **f** The expression of senescence-related secretory phenotypic genes in the liver of doxorubicin-treated mice ($n = 5$/group). **g** The expression of PXR downstream genes in the liver of doxorubicin-treated mice. β-Actin was used as an internal control ($n = 5$/group). **h** Cardiac fibrosis induced by doxorubicin administration, determined by Masson's trichrome staining. **i** The quantitative analysis of fibrosis area in (**h**), ($n = 4$/group). The mice were treated with doxorubicin (5 mg/kg, three times a week) for 2 weeks and nomilin (50 mg/kg/day for 4 weeks). The data were shown as mean ± SEM. $p$-values were determined by one-way ANOVA test (**b, c, e–g, i**). ### $p < 0.001$ *vs* the control group; *** $p < 0.001$ *vs* the doxorubicin group. Dox doxorubicin, NML nomilin.

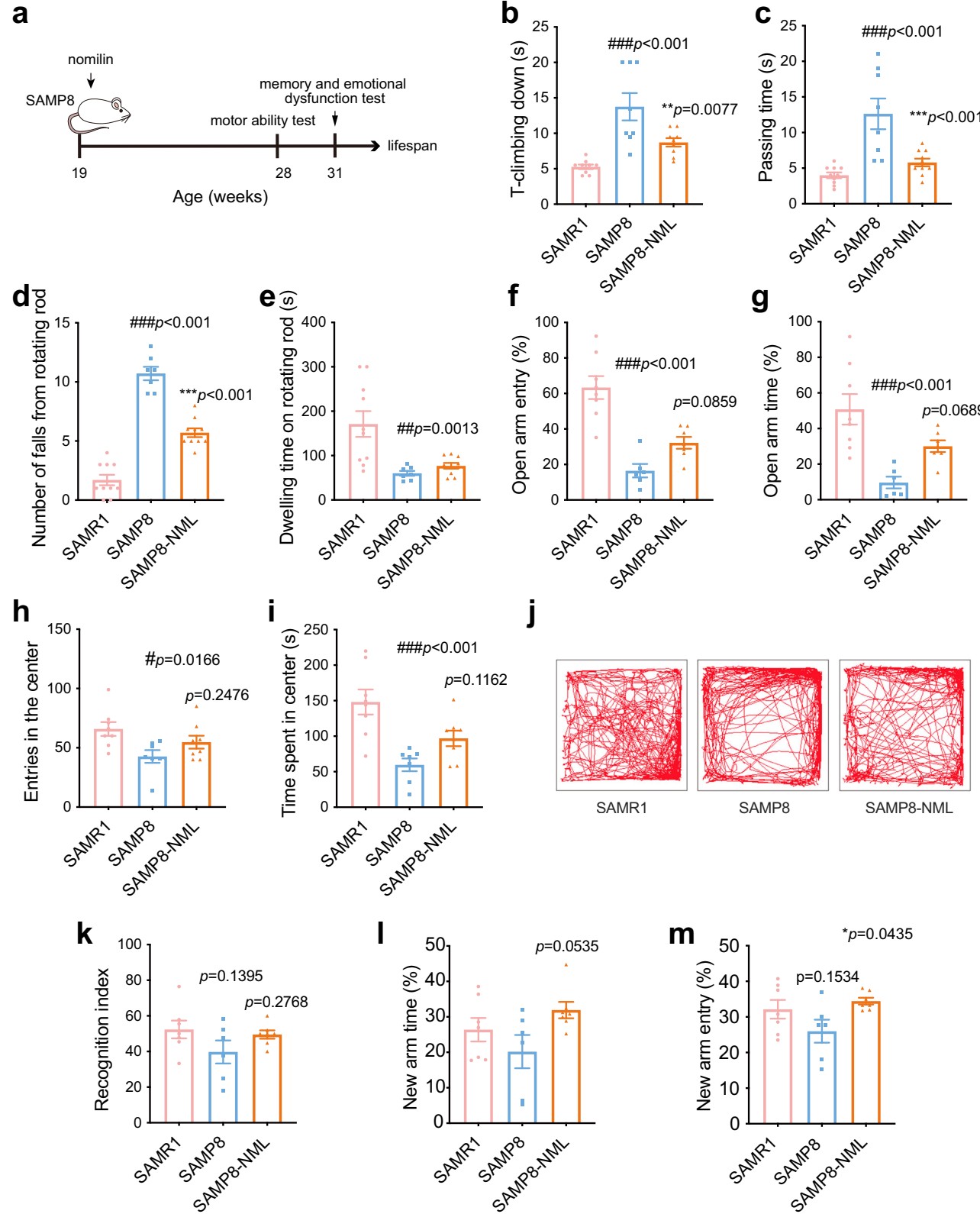

time in the beam balance test (Fig. 8b, c), whereas nomilin treatment reduced the times of T-climbing and passing time (Fig. 8b, c), suggesting that nomilin could improve physical conditions in doxorubicin-induced senescence mice. Next, we assayed whether liver function was also improved by nomilin. In agreement with previous reports, doxorubicin increased the inflammatory cell infiltration in the liver (Fig. 8d, e), serum levels of aspartate aminotransferase (AST) and

alanine transaminase (ALT), indicators of liver damage (Fig. S8a, b). Interestingly, nomilin counteracted the inflammatory cell infiltration in the liver and the increase of AST and ALT (Fig. 8d, e, Fig. S8a, b) in doxorubicin-treated mice. Meanwhile, the expression of senescence-related secretory phenotypic genes *Nlrp3, Tnfα, Il-6, Il-1β, Mcp1, p16^{INK4A}* was increased in the liver of doxorubicin-induced mice, while nomilin treatment downregulated the mRNA levels of *Nlrp3* and *Il-6* (Fig. 8f).

**Fig. 9 | Effects of nomilin on healthspan in SAMP8 mice. a** Timeline for drug treatment and behaviour test. **b** T-climbing down in pole test ($n = 8$ for SAMP8, $n = 9$ for SAMR1 and SAMP8-NML). **c** Passing time in beam balance test ($n = 8$ for SAMP8, $n = 10$ for SAMR1 and SAMP8-NML). **d** Number of falls from rotating rod ($n = 7$ for SAMP8, $n = 10$ for SAMR1 and SAMP8-NML). **e** Dwelling time on rotating rod ($n = 7$ for SAMP8, $n = 10$ for SAMR1 and SAMP8-NML). Proportion of times entering the open arm (**f**, $n = 8$ for SAMR1, $n = 6$ for SAMP8, $n = 7$ for SAMP8-NML) and Proportion of exploration time in the open arm (**g**, $n = 8$ for SAMR1, $n = 6$ for SAMP8, $n = 7$ for SAMP8-NML) in elevated-plus maze. Entries in the centre (**h**, $n = 7$ for SAMP8, $n = 8$ for SAMR1 and SAMP8-NML) and time spent in centre (**i**, $n = 7$ for SAMP8, $n = 8$ for SAMR1 and SAMP8-NML) in open field. **j** Trajectory in open field. **k** Recognition index of mice in novel object recognition test ($n = 6$ for SAMP8, $n = 7$ for SAMR1 and SAMP8-NML). **l** Proportion of time exploring the new arm in Y maze ($n = 6$ for SAMP8, $n = 7$ for SAMR1 and SAMP8-NML). **m** Proportion of times entering the new arm in the Y maze ($n = 6$ for SAMP8, $n = 7$ for SAMR1 and SAMP8-NML). The data were shown as mean ± SEM. $p$-values were determined by one-way ANOVA test (**b–i**, **k–m**). ###$p < 0.001$ vs the SAMR1 group; ***$p < 0.001$ vs the SAMP8 group.

Similarly, the expression levels of mPXR downstream genes *Cyp3a11*, *Por*, *Gsta1/2* and *Mdr3* in the liver were increased by nomilin intervention, indicating that the upregulation of detoxification by nomilin may protect mice from doxorubicin-induced damage (Fig. 8g). Doxorubicin also induced myocardial atrophy and collagen deposition, the markers of fibrosis, in the heart of mice (Fig. 8h, i). Nomilin treatment attenuated cardiomyopathy by reducing cardiac atrophy and fibrosis areas in the heart (Fig. 8h, i). Taken together, these results indicate that nomilin may improve hepatic, cardiac senescence via counteracting toxicity in doxorubicin-induced aged mice. The data further support that nomilin may have a detoxification function.

### Nomilin extends healthspan in SAMP8 mice

The senescence-accelerated mouse is an accelerated ageing model used in gerontological research because of its accelerated senescence and various spontaneous pathobiological phenotypes[76,77]. Here, we chose senescence accelerated mice prone 8 (SAMP8) mouse as an ageing model and senescence-accelerated mouse resistant 1 (SAMR1) mouse as a normal control to investigate the effects of nomilin on accelerated senescence. SAMP8 mice exhibited significant motor impairment, as evidenced by a significant increase in T-climbing, passing time and the number of falls from the rod, and a decrease in the time dwelling on the rod compared to SAMR1 mice. However, these parameters in nomilin-treated SAMP8 mice were reversed (Fig. 9b–e). Previous studies have reported emotional disorders and memory deficits in SAMP8 mice[76,78,79]. In this study, we evaluated anxiety-like behaviour using elevated plus maze and open field tests. Results showed that SAMP8 mice displayed significant anxiety-like behaviour, as indicated by a decrease in the percentage of time spent and the number of entries into the open arms compared to SAMR1 mice, which was consistent with previous results (Fig. 9f, g). In contrast, the nomilin intervention decreased the anxiety-like behaviour of SAMP8 mice (Fig. 9f, g). Similarly, the open field test showed that SAMP8 mice exhibited less exploration of the central area compared to SAMR1 mice, while nomilin-treated mice showed an increase tendency to explore the central region (Fig. 9h, i, j). The novel object recognition experiment was carried out to assess learning and memory abilities of the mice. SAMR1 mice showed a stronger interest in the new object than in the old object, while the recognition index of SAMP8 mice decreased. After nomilin intervention, the mice showed an increase tendency in their ability to recognize new objects (Fig. 9k). And Y-maze test showed that decrease tendency in the percentage of exploration time and number of entries into the new arm in SAMP8 mice when compared to those in SAMR1 mice, whereas nomilin-treated mice increased the number of entries into the new arm (Fig. 9l, m), suggesting that spatial memory impairment in SAMP8 mice was improved by nomilin treatment. Overall, these findings suggest that nomilin may improve age-related disorders such as motor impairments, anxiety-like behaviour, and memory deficits in SAMP8 mice.

### Nomilin activates mPXR and induces a longevity gene signature in mice

Next, we investigated whether nomilin could protect the liver against damage from cholestatic hepatotoxicity through detoxifying toxic bile acids. We supplemented bile duct-ligated (BDL) mice with nomilin and measured their levels of liver damage, since PXR agonist pregnane-16α-carbonitrile (PCN) has been reported to relieve liver damage in this mouse model[80,81]. Our histological assay confirmed that BDL induced severe diffused vacuolization, inflammatory cell infiltration and hepatic parenchymal necrosis in the mouse livers. Notably, similar to PCN (Fig. 10a), nomilin significantly ameliorated the inflammation, fibrosis and necrosis of the liver in WT BDL mice, but not in mPXR knockout BDL mice (Fig. 10a), indicating that nomilin does activate mPXR in vivo in mammals. Moreover, serum biochemical indices showed that nomilin decreased serum ALT and AST levels under BDL surgery, while nomilin showed no effect on either ALT or AST in normal control mice (Fig. 10b, Sham *v.s.* Sham + N), further confirming that nomilin may protect the liver from damage due to BDL injury (Fig. 10b, BDL *v.s.* BDL + N), without toxicity in mice.

Gene expression analysis by transcriptome sequencing (RNA-seq) also confirmed that reported mPXR-induced genes[20,82] are upregulated in the liver of nomilin-treated mice. Specifically, among the 193 genes upregulated by nomilin, at least 27 genes were mPXR downstream targets identified by previous studies (Fig. 10c, d). These genes are involved in drug and toxin metabolism in the mouse liver, which may explain why the liver damage of BDL mice was significantly attenuated under nomilin treatment (Fig. 10d).

It has been reported that most longevity interventions induce common gene expression signatures in the liver of mice, which could be used to predict the lifespan-extension effect of new candidate compounds[19]. For example, the transcript levels of genes coding for ribosomal proteins, oxidative phosphorylation, drug and xenobiotic metabolism-cytochrome P450 enzymes, glutathione metabolism, tricarboxylic acid cycle, amino acid metabolism, age-related neurodegenerative diseases, complement and coagulation cascades, fatty acid oxidation, steroid and retinol metabolism and the peroxisome proliferators-activated receptor (PPAR) pathway were upregulated in the liver of long-lived mice[19]. We further characterised the molecular function and biological processes enriched in the samples using gene annotation. Both Gene Set Enrichment Analyses (KEGG) terms and Gene Ontology (GO) revealed that many top gene categories and metabolic pathways were highly similar to those of most longevity interventions (Fig. 10e–g), indicating that nomilin may share a common molecular pathway for regulating longevity with most lifespan-extending interventions.

## Discussion

Various toxins in the environment are risk factors for human health, and are linked to many age-related diseases such as Alzheimer's disease and Parkinson's disease[83,84]. The increase of detoxification gene expression is a common transcriptomic signature in long-lived worms, flies and rodents, suggesting that xenobiotic detoxification may be linked with longevity-promotion. Nuclear receptors have been identified as regulators of healthy ageing. In mammals, PXR is a major transcription factor for regulating the expression of phase I–III drug metabolising/xenobiotic detoxifying genes. Although many studies have shown that PXR may show cross-talk with longevity signalling[20], it is unknown whether targeting PXR plays a role in ageing inhibition. In the present study, we demonstrated that a component present in citrus fruits, nomilin, is a PXR agonist and extends lifespan and

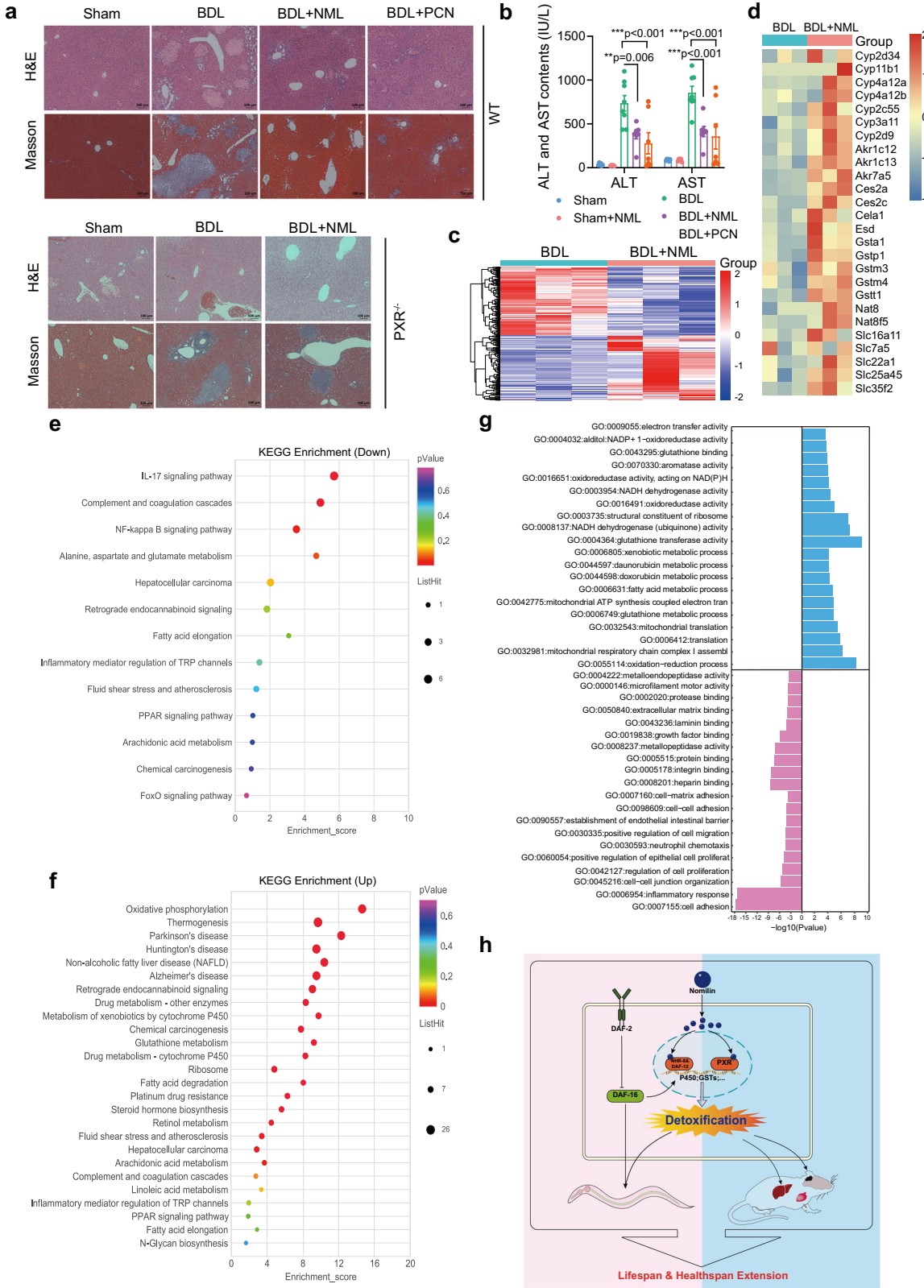

healthspan. Meanwhile, nomilin activates the expression of many phase I–III enzymes and efflux transporters in mice and *C. elegans*, which is correlated with IIS signalling (Fig. 10h). These data are in line with the results of most longevity interventions, such as caloric restriction, which upregulates most phase-I/II enzymes and phase-III efflux transporters in the livers of male mice[85], further supporting that detoxification may be a mechanism for longevity-promotion. Our

results show that PXR may have physiological functions in ageing in addition to drug metabolism.

Interestingly, citrus and grapefruit juices may contain a large amount of nomilin or its precursors, which could be hydrolysed in the liver and by the intestinal flora[26]; however, most of them would be removed in the "debittering" processing in the orange juice industry because of their bitter taste[86]. We showed that nomilin-treatments did

**Fig. 10 | Nomilin protects BDL-induced liver cholestatic injury through mPXR and upregulates longevity related genes in mice. a** Pictures showing H&E or Mason staining of liver sections in BDL mice with or without PCN and nomilin supplementation. Nomilin effectively attenuated the BDL-induced liver damage in WT (upper), but not in PXR$^{-/-}$ mice (bottom). **b** Bar-graphs showing the levels of serum ALT and AST in sham or BDL mice ($n = 7$ for BDL + NML and BDL + PCN, $n = 8$ for BDL). The data are shown as mean ± SEM. *p*-values were determined by one-way ANOVA test. ****p* < 0.001 *vs* BDL group. **c, d** The RNA-seq hierarchical clustering heatmap showing differentially expressed genes (**c**) and genes in the detoxification process (**d**) from control (BDL) and nomilin-treated (BDL + NML) mouse liver under BDL surgery. **e, f** A chart showing the downregulated (**e**) and up-regulated (**f**) differentially expressed genes of KEGG pathways compared with the control group in both nomilin-treated and long lifespan mice. **g** A chart showing the top 20 up-regulated (blue) and downregulated (pink) molecular functions and biological processes (GO). **h** A diagram depicting the effects of nomilin on longevity through the activation of nuclear hormone receptors and detoxification signalling in *C. elegans* and mice.

not change the body weight and food consumption of the mice (Supplementary Fig. S9a-h), indicating that nomilin may be a safe component. Therefore, our results suggest that a revisit of the "deb-ittering" process may be needed given the potential beneficial function of nomilin. Notably, as a xenobiotic-sensor to protect the body from endo/xenobiotics by detoxification of toxins, PXR was originally characterised as a regulator of drug metabolism[87]. As a PXR agonist, nomilin may accelerate drug metabolism and attenuate the efficiency of therapy. Whether the consumption of nomilin-containing citrus fruits and juices change drug metabolism needs to be investigated.

In conclusion, we found that nomilin extends the lifespan and healthspan in *C. elegans* and mice, and regulates the gene expression of detoxification enzymes through the activation of nuclear hormone receptors. The detoxification function of nomilin is probably linked to IIS longevity signalling. Our data suggest that targeting PXR maybe a feasible strategy for longevity and health promotion.

## Methods

### Chemicals
Nomilin and limonin (purity > 99.8%) were obtained from Pusi Biotech (Chengdu, China). Nomilin identity was confirmed by a mass spectrometry assay (see Supplementary Methods). Chloroquine (Yuanye, Shanghai, China), colchicine (Yuanye, Shanghai, China), paraquat (Thermo Fisher Scientific, Waltham, USA), methylmercury chloride (MeHgCl, Dr. Ehrenstorfer GmbH, Augsburg, German), PCN (GLPBIO, Montclair, USA) and doxorubicin hydrochloride (Yuanye, Shanghai, China) were commercially available.

### *C. elegans* strains and maintenance
The following *C. elegans* strains were used in this study: N2: Wild-type Bristol isolate, CB1370: *daf-2 (e1370)*, CF1038: *daf-16 (mu86)*, MAH97: *muIs109 [daf-16p::GFP::DAF-16 cDNA + odr-1p::RFP]*, VC199: *sir-2.1* (ok434), VC222: *raga-1* (ok386), RW12220: *pha-4 (st12220[pha-4::TY1::EGFP::3xFLAG])*, DR2281: *daf-9* (m540), AA86: *daf-12* (rh61rh411). These were obtained from the CGC (Caenorhabditis Genetics Center), which is funded by the NIH National Center for Research Resources (NCRR). *nhr-8 (tm1800)* was obtained from National BioResource Project (Tokyo, Japan). The worms were cultured on nematode growth medium (NGM) agar plates seeded with live bacteria at 20 °C (E. *coli*, strain OP50) as food source, according to standard protocols[88,89].

### Lifespan experiments
Synchronised L1 worms were cultivated on standard NGM plates at 20 °C for about 3 days. Then, L4 adults were transferred to plates (30 worms per plate), fed with 0, 25, 50 100 μM nomilin and limonin or DMSO (0.1%) solvent control mixed with OP50, respectively. Worms were judged as dead when they did not respond to repeated prodding with a pick and had no pharynx pumping. Dead worms were counted daily. Worms that crawled off plates or bagging were excluded.

Heat-killed OP50 were prepared with a 20× concentrate for 1 h at 75 °C, as previously described[90].

For RNA interference (RNAi) lifespan experiments, synchronised L1 worms were fed with *E. coli* (HT115) containing an empty control vector (L4440) until L4, then transferred to plates where they were fed with *daf-12, nhr-8, cyp35a3, gst-4, pgp-3 or pgp-14* RNAi constructs with or without nomilin (50 μM), using L4440 and DMSO (0.1%) as controls on NGM containing isopropyl-beta-D-thiogalactopyranoside (IPTG, 1 mg/ml) and ampicillin (50 μg/ml). All RNAi constructs were obtained from the Ahringer RNAi library and grown at 37 °C overnight in LB containing ampicillin (50 μg/ml) after sequence verification.

### DAF-16:GFP translocation experiments
muIs109 [*Pdaf-16::gfp::daf-16; Podr-1::rfp*][91], *Pdaf-16::gfp::daf-16; Podr-1::rfp; nhr-8 (1800)*, *Pdaf-16::gfp::daf-16;* and *Podr-1::rfp; daf-12 (rh61rh411)* worms were treated with nomilin from L1 to L4, anaesthetised with 100 mM NaN$_3$ and mounted on 2% agarose pads. The GFP fluorescent signals of DAF-16 localisation were examined in 10 animals per condition and captured by a confocal microscope (SP-8 Leica, Germany). Images were acquired with a digital camera. The number of GFP-positive nuclei of each worm was calculated.

### Cross strategy
To obtain a homozygous strain of *Pdaf-16::gfp::daf-16; Podr-1::rfp; nhr-8 (1800)* and *Pdaf-16::gfp::daf-16; Podr-1::rfp; daf-12 (rh61rh411)*, muIs109 males were mated with daf-12 (rh61rh411) or nhr-8 (tm1800) hermaphrodites (P0). F1 worms carrying the RFP fluorescence were considered as cross-progeny and singled into 10 35 mm NGM plates. F2 with all progeny carrying the RFP fluorescence were considered as muIs109 homozygous and singled for *nhr-8 (tm1800)* or *daf-12 (rh61rh411)* genotyping. *nhr-8(-)* homozygous worms were identified by PCR with the primers nhr-8 F (catttatacttctaaaccaacaattgt), nhr-8 D (ccggataatttcattgaaacttact), and nhr-8 R (ggtacatatcacaggttatcgaga). *daf-12(-)* homozygous worms were identified by sequencing using daf-12 F (attgtatttcagggtatcatggatc) and daf-12 R (ggtgataaatgtggctgttgatta).

### Heat stress and oxidative stress experiments
In stress resistance assays, the number of surviving worms was monitored following exposure to the indicated stressor. For heat shock experiments, L4 phase N2 worms were treated with nomilin for 10 days, and then the worms were placed at 35 °C for 12 h. Every 2 h the worms were observed for survival. The experiments were repeated three times. For oxidative stress resistance experiments, the L4 worms were placed on 0.05% H$_2$O$_2$ NGM agar plates for 12 h at 20 °C. Every 2 h the worms were observed for survival.

### Dauer induction assay
The dauer induction by high-density growth was performed according a previously reported method[92], except that 50 μl *E. coli* OP50 with 10% DMSO or 50 μM nomilin was seeded on the egg white plates (about 1.5–6 × 10$^4$ eggs/plate). The dauer induction in the *daf-2(e1370)* mutation was carried out as a standard protocol in WormBook[93]. Briefly, the worms were maintained at 15.0 °C on standard NGM plates, and allowed gravid adult hermaphrodites to lay eggs for several hours on 3.5 mm NGM plates that seeding with 100 μl *E. coli* OP50 (with 10% DMSO, or 50 μM NML) at 20.0 °C, then removed adults when about 100–200 eggs were laid on the plates, and shifted the plates to a incubator for the dauer formation at 22.0/23.5 °C for 68–80 h. Then, 1 ml of 1% sodium dodecyl sulphate was added to the plates and incubated for 30 min to count dauers (survivors).

## Cell cultures and reporter assays

HEK 293 T cells (CRL-11268, ATCC) were maintained in Dulbecco's modified eagle medium with 10% foetal bovine serum (FBS, Hyclone, Logan, UT, USA). For reporter assays, the expression plasmid pSG5-hPXR/CYP3A4-Luc, pcDNA3.1-hCAR/CYP3A4-Luc, pCMXGal-hPPARα, β, γ LBD, LXRα/β LBD, and the Gal4 reporter vector MH1004-TK-Luc were cotransfected with pREP7. For transfection, each well contained 100 ng of total plasmids and 0.2 μl of FuGENEHD transfection reagent (Roche, Germany) for 24 h. Then, nomilin and hPXR, hCAR, PPARα, β, γ, LXRα, β, and FXR agonist (Rifampicin, CICTO, fenofibric acid, GW4064, pioglitazone T0901317 and GW4064, respectively, Sigma Aldrich, St. Louis, MO, USA) were added to fresh media and incubated for another 24 h. The luciferase activity was measured using the Dual Luciferase Reporter Assay System (Promega, USA), and the transfection efficiencies were normalised according to Renilla luciferase activity.

## TR-FRET assay

Time-resolved fluorescence resonance energy transfer (TR-FRET) was performed using a LanthaScreen™ TR-FRET PXR (SXR) competitive binding assay kit according to the manufacturer's protocol (PV4839, Invitrogen, Darmstadt, Germany). Briefly, labelled hPXR-ligand Fluormone™ PXR (SXR) Green (40 nM), hPXR-LBD-GST (5 nM), goat terbium-anti-GST antibody (5 nM) and nomilin, deacetylnomilin, limonin and/or T0901317 (10 μM) were incubated in assay buffer at room temperature for 2 h, and the 520 and 495 nm fluorescence signal was assayed using a PerkinElmer EnVision Multilabel Reader. The 520/495 value was calculated by subtracting the background TR-FRET ratio. The dissociation constant ($K_d$) was fitted into a one-site total binding saturation equation in GraphPad Prism software. $IC_{50}$ values were determined using log (inhibitor) vs. response - Variable slope model fit by GraphPad Prism software according to the previous method[94]. All experiments were performed in triplicate.

## Protein expression and purification

A cDNA encoding a PXR$^{LBD}$-NCOA1$^{676-700}$ fusion protein comprised of residues 130–432 of hPXR (Uniprot ID O75469-1) with its C-terminal linked to residues 676–700 of nuclear receptor coactivator 1 (Uniprot ID Q15788-1) and spaced with -Ser-Ser-Ser-Gly-Gly-Thr- was synthesised and cloned into a plasmid modified from pFastBac Dual (Invitrogen), with a C-terminal TEV protease recognition site and 6 × polyhistidine affinity tag. The recombinant baculovirus encoding the hPXR$^{LBD}$-NCOA1$^{676-700}$ fusion protein was generated and used to infect the *Spodoptera frugiperda* cell line Sf9 for overexpression. The Sf9 cells were harvested 48–72 h after infection and collected by centrifugation (1500 g, 15 min, 20 °C). To lyse the cells, the pellets were re-suspended with ice-cooled buffer containing 150 mM NaCl, 20 mM HEPES, pH 7.5, 0.1 mg/ml DNase I, 2 mM $MgCl_2$, 1 mM TCEP and protease inhibitor cocktail, and subjected to sonication lysis. The cell lysate was clarified by centrifugation at 46,000 g for 45 min and the supernatant was subjected to immobilised metal affinity chromatography (IMAC) with Talon Metal Affinity Resin (Clontech). After the removal of the tag with tobacco etch virus protease (recombinant protein with His-tag), the imidazole in eluate was removed by dialysis against buffer containing 150 mM NaCl, 20 mM HEPES, pH 7.5, 10% (v/v) glycerol and 1 mM TCEP, and then subjected to IMAC to remove tobacco etch virus protease. The flow-through fraction containing the hPXR$^{LBD}$-NCOA1$^{676-700}$ fusion protein was further isolated using size-exclusion chromatography with a Superdex 200 Increase 10/300 GL column (GE Health Sciences), equilibrated in 150 mM NaCl, 20 mM HEPES and 1 mM TCEP. The peak fractions containing the hPXR$^{LBD}$-NCOA1$^{676-700}$ fusion protein was pooled and supplemented with nomilin to a final concentration of 1 mM, followed by concentrating to ~7 mg/ml for crystallization trials.

## Crystallization and X-ray data collection/processing

The hPXR$^{LBD}$-NCOA1$^{676-700}$ was mixed with crystallization solution comprising 10% (v/v) 2-propanol, 100 mM imidazole/hydrochloric acid pH 8.0 at an initial ratio of 1:1. Crystals were grown at 4 °C using the sitting drop vapour diffusion method and were cryoprotected by dipping in crystallization solution supplemented with 20% glycerol and then flash-freezing in liquid nitrogen. Diffraction data were collected at the Shanghai Synchrotron Radiation Facility (SSRF), beamlines BL18U1 and BL19U1. The data were collected and processed with HKL2000[95]. The crystallographic parameters and data collection statistics are given in Supplementary Table S3. The hPXR structure model with PDB ID 5X0R was used as the search model[56], and the molecular replacement and initial model building were performed in Phenix[96]. Iterative cycles of refinement were carried out using PHENIX and Coot[97]. All structure graphs in this paper were produced using PyMOL (The PyMOL Molecular Graphics System, Version 1.9 Schrödinger, LLC.) and LigPlot$^+$ (LigPlot$^+$ version v1.4.5)[98].

## Generation of hPXR point mutations

hPXR mutations M243Q, M243A, S247R, S247A, W299R, M425Q and M425A were created using the pSG5-hPXR expression plasmid as a template, and PrimeSTAR DNA polymerase was used to amplify the DNA. The primers for PCR are:

M243A (ATG → GCT):
TGCCCCACGCTGCTGACATGTCAACCTACAT;
ATGTCAGCAGCGTGGGGCAGCAGGGAGAAGAT.

M243Q (ATG → CAA):
TGCCCCACCAAGCTGACATGTCAACCTACAT;
ATGTCAGCTTGGTGGGGCAGCAGGGAGAAGA.

S247A (TCA → GCT)
CTGACATGGCTACCTACATGTTCAAAGGCAT;
ATGTAGGTAGCCATGTCAGCCATGTGGGGCA.

S247R (TCA → AGA):
CTGACATGAGAACCTACATGTTCAAAGGCAT;
ATGTAGGTTCTCATGTCAGCCATGTGGGGCA.

W299R (TGG → AGA):
CTGGAACCAGAGAGTGTGGCCGGCTGTCCTA;
CCACACTCTCTGGTTCCAGTCTCCGCGTTGA.

M425A (ATG → GCT):
TACGCCCCTCGCTCAGGAGTTGTTCGGCATCACA;
AACTCCTGAGCGAGGGGCGTAGCAAAGGGGTGTA.

M425Q (ATG → CAA):
TACGCCCCTCCAACAGGAGTTGTTCGGCATCACA;
AACAACTCCTGTTGGAGGGGCGTAGCAAAGGGGT.

All mutations were confirmed by sequence analysis. The reporter gene assay was carried out as described above.

## RNA extraction and real time RT-PCR

For the gene expression assay, 50 μM nomilin-treated N2, *daf-12(-)* and *nhr-8(-)* or untreated L4 worms were subjected to quantitative real-time PCR. The total RNA was extracted from about 1000 worms using the TRIzol reagent (Sangon Biotech, Shanghai, China) according to the manufacturer's instructions. The residual DNA was removed using gDNA wiper mix, and 1 μg of total RNA was reverse-transcribed to complementary DNA using HiScript II qRT SuperMix II (Sangon Biotech, Shanghai, China). Quantitative real time PCR was performed using the ABI StepOnePlus Real Time PCR system (Applied Biosystems, Foster City, CA, USA) using SYBR Green PCR Master Mix (Sangon Biotech, Shanghai, China). The results were analysed with β-actin as the internal control. Sequences for primers are listed in Supplementary Table 5.

For mouse experiments, 10 mg of tissues were used to extract total RNA and quantitative PCR was performed as described above. The primer sequences are listed in Supplementary Table 6.

## Western blot analysis

For total protein extraction, the liver tissues were homogenised in sample buffer and boiled for 5 min. The samples were separated using 10% SDS–PAGE, transferred to a PVDF membrane, and blocked with 5% bovine serum albumin at room temperature for 2 h. Then, the blots were incubated with polyantibodies against Gsta, Cyp3a11, Cyp51a1 and GAPDH (ProteinTech, Rosemont, USA) at 4 °C for 12 h. The membrane was washed and incubated with secondary antibody for 2 h at room temperature. The signals were detected and analysed using an Odyssey Two-Colour Infrared Imaging System (LI-COR Biosci- ences, Lincoln, NE, USA). GAPDH was assayed as a loading control.

## Generation of transgenic worms

Human PXR and the hPXR$^{S247R}$ cDNA sequence were cloned and driven by Prpl-28 promoters and injected into the wild-type N2, nrh-8 (-) and daf-12 (-) mutant lines with the Pmyo-2::RFP co-injection marker under the IM 300 Microinjector (NARISHIGE, Japan). pSM delta vectors were injected into the three lines as the vehicle controls.

The fluorescence-marked transgenic strains pSM; Pmyo-2::rfp, pSM; Pmyo-2::rfp; nrh-8(tm1800), Pmyo-2::rfp; daf-12 (rh61rh411), Prpl-28:hPXR; Pmyo::rfp, Prpl-28:hPXR; Pmyo::rfp; nrh-8 (tm1800), Prpl-28:hPXR; Pmyo::rfp; daf-12 (rh61rh411), Prpl-28:hPXR$^{S247R}$; Pmyo::rfp, Prpl-28: hPXR$^{S247R}$; Pmyo::rfp; nrh-8 (tm1800), Prpl-28: hPXR$^{S247R}$; and Pmyo::rfp; daf-12 (rh61rh411) were generated for life-span experiments with DMSO and nomilin treatment.

## Detoxification assay

Detoxification assays were performed in 12-well polystyrene tissue culture plates. Concentrated stock solutions of chloroquine, colchicine, paraquat and MeHgCl at 50 mM were prepared in complete S medium, and filtered through 0.22 μm nitrocellulose filters. The working solutions were diluted to the indicated concentrations using S buffer (chloroquine, colchicine and MeHgCl) or M9 buffer (paraquat). OP50 was added to the working solutions. Serial dilutions were made in bulk and aliquoted into individual wells (1 ml per well). The synchronous N2, nrh-8 (-) and daf-12 (-) worms were treated with nomilin at concentrations of 0, 6, 12, 25, 50 and 100 μM from the L1 to L4 stage. Then, the worms were transferred to S buffer or M9 buffer containing the same concentration of nomilin plus toxins (4 mM chloroquine and colchicine, 2 μM MeHgCl in S buffer, and 100 mM paraquat in M9 buffer) in 96-well plates, each containing five worms. The worms were cultured and monitored under a stereo microscope at the indicated time points.

For the detoxification lifespan assay, adult (Day 10) N2, daf-2, daf-16 and daf-2 with nrh-8 or daf-12 RNAi (from L4) were challenged with paraquat (100, 200 mM) in M9 or MeHgCl (1, 2 μM) in S buffer for 24 h. The death rate was recorded at 1, 2, 4, 6, 8 and 24 h. Meanwhile, nomilin (12, 50 μM)-treated N2 (from L4 to Day 10) were also picked into paraquat or MeHgCl buffer containing nomilin (12, 50 μM).

## D-galactose induced senescence in mice

Sixty 8-week-old female and male C57BL/6 mice were obtained from Shanghai Laboratory Animal Center of Chinese Academy of Science (Shanghai, China). All mice were housed under a 12 h light/dark cycle in a room with controlled temperature (22 ± 1 °C). All procedures were approved by the Experimental Animal Ethical Committee at Shanghai University of Traditional Chinese Medicine (PZSHUTCM191122007). After 1 week of adaptive feeding, the mice were randomly divided into three groups (20 mice per group, half male and half female): control group (0.9% saline + normal diet), model group (125 mg/kg/day D-galactose + normal diet) and nomilin intervention group (125 mg/kg/day D-galactose + 50 mg/kg nomilin mixed into diet). Mice were injected subcutaneously with 0.9% saline or 125 mg/kg D-galactose daily for 7 weeks. Behaviour assessments were performed after 6 weeks of treatment, and the mice were sacrificed after anaesthetised

with 20% urethane (Sinopharm Chemical Reagent Co., Shanghai, China) for further study.

PXR null knockout mice (C57BL/6N-Nr1i2$^{em1Cya}$) were purchased from Cyagen Biosciences (Suzhou, Jiangsu, China). The detailed information on this mouse can be seen at https://www.cyagen.com/cn/zh-cn/sperm-bank-live/18171. Twenty-four 8–12-week-old female and male PXR$^{-/-}$ mice were divided into the control group, D-galactose group and D-galactose+nomilin group (n = 8 in each group). D-galactose induced senescence and nomilin treatment was performed as indicated above.

## Doxorubicin induced accelerated ageing in mice

To induce senescence in the model, male C57BL/6 mice (6–8 weeks) were intraperitoneally injected with doxorubicin at 5 mg/kg three times weekly for 2 weeks. The control group mice were administered an equivalent volume of saline. For lifespan experiments, the mice were contained to receive doxorubicin (5 mg/kg three times per week) and nomilin (50 mg/kg/day) until all mice died. For the heath-span experiments, the mice were given doxorubicin (5 mg/kg three times per week) for 2 weeks and nomilin (50 mg/kg/day) by oral gavage for 4 weeks. The mice were anaesthetised with 20% urethane (Sino-pharm Chemical Reagent Co., Shanghai, China), and blood and tissues were harvested and analysed.

## SAMP8 mice

Nineteen male SAMP8 mice and ten male SAMR1 mice were obtained from Beijing HFK Bioscience, China. SAMP8 mice were divided into two groups randomly according to their body weight. NML was mixed into food (at dose of 40 mg/100 g diet) to feed mice from 19-week-age old. And then these mice were taken to test their behaviour at the indicated timeline in Fig. 9a. Body weight and food intake were recorded every 2 days. The animal protocols were approved by the Experimental Animal Ethical Committee at Shanghai University of Traditional Chinese Medicine (PZSHUTCM2212020004).

## Behaviour assessments of mice

**Pole test**. The pole test uses a device consisting of a wooden stick (diameter 1 cm, height 52 cm) and a wooden ball (diameter 2.5 cm) at the top of the stick. The device was wrapped with medical tape to prevent mice from slipping. Mice were placed head down and hind paws were placed on the ball in order to record the time of climbing down the stick. In the 6th week of treatment, each mouse was allowed to perform two trials and the average value was used for statistical purposes. In this process, T-climbing down would be recorded as 20 s when the mouse took >20 s to climb down to the cage from the pole.

**Balance beam test**. To measure coordination and balance, a cylindrical wooden stick (1 cm in diameter and 50 cm in length) wrapped with medical tape was placed above the cage, and the cage was covered with a layer of bedding to prevent mice from being injured. The mice were placed on one side of the stick, and the time for the mice to reach the other end of the wooden stick was recorded (the front paws touching the edge of the cage was considered as successful arrival). Each mouse was given two trials and the interval time between trails was 30 min. When the mouse walked from one side of the stick to the other for >30 s, the passing time was recorded as 30 s.

**Rotarod test**. The mice were placed on a rotarod apparatus (Shanghai Bio-will Co., Ltd.) to examine motor dysfunction associated with neu-rological impairment. Before testing, the mice were put on the rod for 3 min at 5 rpm to acclimate to the device. Two hours later, the rod speed was accelerated from 5 to 40 rpm and the mice was put on for 5 min. The time of the mice fell from the rod and the number of times it fell from the rod within 5 min were recorded. Two trials were performed per mouse with a 30 min interval between trials.

**Gait analysis.** Gait analysis is used to assess neurological and neuromuscular functions by detecting walking parameters in freely moving mice[99]. Before the test, the mice were place in the testing room without interference and light for 1 h to adapt. Each mouse was put on the left initial terminal and trained to voluntarily walk along the glass track to the other side. Based on optical technology from the Catwalk Automated Gait Analysis System (Noldus Information Technology, Wageningen, Netherlands), three correct runs were recorded for each animal and the associated gait parameters were analysed using CatWalk XT version 10.6.

**8-arm maze.** The 8-armed maze was used to evaluate the learning and memory ability of mice. The basic principle of this experiment is that controlling mouse explores the arms of maze driven by food. After a period of training, animals can remember the spatial position of food in the maze. The experiment used an 8-arm radial maze, each arm is 50 cm long, 7 cm wide, and 11 cm high. In the center, there is a circular platform with a diameter of 25 cm, leading to eight arms. Mice fasted for 12 h before training, and each mouse was given 2–3 g food per day throughout the experiment to stay hungry. On the first day, about 10 mg of bread was placed at the end of each arm, and three mice were placed on the central plate. The door to each arm was opened and the mice were allowed to explore the maze for 10 min. The training was repeated the next day. Again, bread was placed at the end of each arm, but on the third and 4th day of training, only one mouse was placed in the maze for 5 min each time. After a day off, only two randomly selected arms had bread at the ends. The end of the test signal is that two pieces of bread have been eaten or the exploration time has reached 5 min. This training lasted for 5 days. On the 11th day, performance rate (P) and exploration time were recorded to reflect the short-term memory ability of mice[100]. The formula for calculating the performance rate is listed below, where n refers to the total number of times entering the arm, reference memory error (RME) refers to the number of visits to an arm without food, working memory error (WME) refers to the number of visits to a previously visited arm. After 3 days of rest, the test was repeated and the above indicators were recorded to show the long-term memory ability of the mice.

$$P = \frac{n - (RME + WME)}{n} \times 100\% \qquad (1)$$

**Elevated plus-maze test.** The elevated plus-maze consists of crisscrossing open and closed arms, of which the closed arm is surrounded by 15 cm high wall. The entire experimental setup is 1 m above the ground. The experiment examined the anxious behaviour of rodents due to their dislike of open field and heights. The mouse was placed in the central area of the maze, with its back to the experimenter and facing the open arm. And it was allowed to move freely through the maze for 5 min. The proportion of times the mouse entered the open arm and the proportion of time spent in the open arm were calculated.

**Open field test.** The open field test was used to evaluate the inquiry behaviour and the tension of mice in an unfamiliar environment. Mice were placed on a square open field of 50 cm × 50 cm × 50 cm, and the time and number of mouse exploration in the central region were recorded to assess the anxiety behaviour.

**Y-maze.** The Y-maze is a classical behavioural test used to measure spatial memory in mice[101]. The device consists of three arms of equal length (30 cm × 5 cm × 15 cm). The learning and memory ability of mice is displayed by detecting the number of times exploring the new arm and the total number of explorations. The gate of new arm was closed during the training period, and each mouse was placed facing the wall in the starting arm. Then it was allowed to explore the maze freely for 10 min and learn to remember the spatial position of the remaining two arms. After an hour break, the gate of the new arm was opened, and the mouse was placed in the starting arm facing the wall, and it was freely explored for 5 min. The entire experiment was recorded with a camera. And frequency enter to each arm and explore time in each arm was counted by the EthoVision XT analysis system. The memory ability is expressed as the proportion of frequency of entering to new arm and proportion of time exploration in new arm.

**Novel objective recognition.** Novel object recognition was used to detect the cognitive abilities of mice based on their nature of being intensely curious about new targets. The device includes a cube field with a side length of 50 cm and three objects of different shapes and colours. During the acclimatization period, two identical objects A were placed in the field, each about 10 cm away from the wall. Then mouse was placed in the field with back to the object and at the same distance from both objects. EthoVision XT analysis system was used to record the exploration time of mice on each object (touching the object with the mouth or nose and approaching the object within a range of about 2–3 cm are considered to be exploring the object). After 10 min exploration, mouse was put back into its cage. And after 1 h of rest, the mouse was placed back in the field. At this point, one of the two objects in the field was replaced with a new object B. And, the recognitive index refers to the proportion of times exploring the new object.

### Bile duct ligation (BDL) experiments
All animal experiments were approved by the Ethical Committee of Shanghai University of Traditional Chinese Medicine (Approval number: PZSHUTCM190609001). Male C57BL/6 mice and PXR$^{-/-}$ mice (9-week-old, body weight >25 g) underwent BDL surgery under 1% pentobarbital anaesthesia according to the previous description, using a sham group as a control[102]. BDL-surgery mice were divided into two groups ($n = 9$–10) 1 day after surgery and treated with nomilin (100 mg/kg), PCN (100 mg/kg) or vehicle (0.5% CMC-Na) orally for 2 weeks. At the end of the experiment, mice were anaesthetised with 20% urethane (Sinopharm Chemical Reagent Co., Shanghai, China) after overnight fasting. Heart blood samples were taken and serum was separated for ALT and AST analysis using an automatic biochemical analyser (Hitachi 7020, Japan).

### RNA sequencing analysis
The liver tissues of the mice were collected for RNA sequencing analysis. Total RNA from BDL and BDL + NML groups was isolated from mouse liver tissue using the TRIzol reagent according to the manufacturer's protocol. RNA purity and quantification were evaluated using the NanoDrop 2000 spectrophotometer (Thermo Scientific, USA). RNA integrity was assessed using the Agilent 2100 Bioanalyzer (Agilent Technologies, Santa Clara, CA, USA). Then, the libraries were constructed using the TruSeq Stranded mRNA LT Sample Prep Kit (Illumina, San Diego, CA, USA) according to the manufacturer's instructions. The transcriptome sequencing and analysis were conducted by OE Biotech Co., Ltd. (Shanghai, China).

The libraries were sequenced on an Illumina HiSeq X Ten platform and 150 bp paired-end reads were generated. Raw data in fastq format were firstly processed using Trimmomatic and the low-quality reads were removed to obtain the clean reads. Then, clean reads for each sample were retained for subsequent analyses. The clean reads were mapped to the Mus musculus genome (GRCm38.p6) using HISAT2. The fragments per kilobase million of each gene was calculated using Cufflinks, and the read counts of each gene were obtained using HTSeq-count. Differential expression analysis was performed using the DESeq (2012) R package. $P$-value < 0.05 and fold change > 1.5 were set as the thresholds for significantly differential expression. Hierarchical cluster analysis of differentially expressed genes (DEGs) was performed to demonstrate the expression pattern of genes in different

groups and samples. GO enrichment and KEGG pathway enrichment analysis of DEGs were performed using R, based on the hypergeometric distribution.

### Statistical analysis

Results of lifespan experiments were analysed using Kaplan-Meier survival analysis and compared among groups, scoring for significance using the log-rank test. The results of survival values following stress conditions were analysed using Student's *t*-test. For Supplementary Table 1, 2 and 4, the average of the mean lifespan, the minimum and the maximum lifespan of a set of independent experiments were calculated and expressed as mean ± SEM. SPSS was used for statistical analysis. A *p*-value of 0.05 or less was considered to be statistically significant.

### Reporting summary

Further information on research design is available in the Nature Portfolio Reporting Summary linked to this article.

## Data availability

Shanghai Synchrotron Radiation Facility (SSRF) beamlines BL18U1 and BL19U1 are used for X-ray crystallography data collection. The coordinates are deposited under the PDB accession code: 7YFK. RNA-sequencing data have been uploaded to the SRA (The Sequence Read Archive) database with accession number PRJNA743088. The rest of data generated in this study are provided in the Supplementary Information/Source Data file. Source data are provided with this paper.

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

## Acknowledgements

This work was supported by the National Natural Science Foundation of China (82073951, 32170837) to C.H. and H.Z., the Science and Technology Commission of Shanghai Municipality (20S11902000, 16PJ1407400) to Y.C. and H.Z., the National Key R&D Program of China (2019YFA0802804, 2021YFA0804701) to H.Z., the Recruitment Program of Global Experts of China (Youth) to H.Z. and the Fok Ying Tung Education Foundation (171036) to S.F. We thank Xiaoming Li, Ziwei Yang and Chengyu Fan from the Molecular Imaging Core Facility (MICF), Ying Xiong, Xiaoyue Ren from the Molecular and Cell Biology Core Facility (MCBCF) at the School of Life Science and Technology, ShanghaiTech University, and Dr. Di Chen from Nanjing University for sharing *C. elegans* strains.

## Author contributions

H.Z., Y.C. and C.H. conceived the study. H.Z., Y.C., C.H. and S.F. designed the experiments. S.F., Y.Y., Z.Z. and M.Z. performed pharmacological studies in animals with assistance from Y.H., D.Y., L.Z., M.F., L.P., J.Y., Y.L. and X.G. S.F., Y.Y., Z.Z. and L.L. performed molecular biology studies. Y.X. performed structural biology experiments. H.G. and C.W. performed the mass spectrometry analysis. S.F., Y.Y., Z.Z., Y.H., D.Y., L.Z., M.F., L.P., J.Y., Y.L. and X.G. analyzed the results. H.Z., Y.C., C.H. and S.F. wrote and edited the manuscript. S.F., C.W., X.W., H.Z., Y.C. and C.H. supervised the project and gave discussion and edited the manuscript. All authors commented on the manuscript.

## Competing interests

The authors declare no competing interests.
