## [Peer Review File · Nature Communications]

Pregnane X receptor agonist nomilin extends lifespan and healthspan in preclinical models through detoxification functionsREVIEWER COMMENTS

Reviewer #1 (Remarks to the Author):

In this paper, Fan et al investigate the properties of the citrus derived compound nomilin (NML) in nematodes and mice for a protective function in health and ageing. NML robustly extends lifespan and appears to do so through IIS pathway. They then focus on detoxification mechanisms and show that the beneficial effects of NML and nuclear DAF-16 localization depend on *nhr-8* and *daf-12*. In fact, some detoxification genes were required for the NML longevity. Elegantly switching to structural analysis, the authors show that NML binds hPXR and use amino acid substitutions to assess NML interaction in its hPXR binding pocket. Next, they ask if hPXR and NML can affect worm survival. Further analyzing NML in mice, the authors show a neuroprotective effect using two paradigms.

This work shows an impressive breadth of methods and reports very interesting data. Some of the conclusions, however, are not fully supported by the data and will require some additional experimentation.

Essential points:

1. To solidify the effect of NML through IIS and the NHRs, it would be important to check for the expression of known DAF-16 target genes. Does NML affect dauer formation?
2. For the longevity assay statistics, it would be important to show the number of censored worms. Also, while the authors use large numbers of worms in each experiment, it would be important to do independent biological repeats of the lifespan assays.
3. To solidify the survival experiments, it would be useful to use limonin as a negative control.
4. I see a few inconsistencies in the lifespan data in Fig 4. The lifespan extension in the WT is smaller than that reported in Fig 1. While I really like the idea of testing hPXR in the worm, I do not see the evidence in the data that it can replace NHR-8 or DAF-12. To look at this, there should be a comparison between the “rescue” strains with WT and mutant hPXR in Table S6.
5. What is the effect of hPXR expression on the target genes of NHR-8 and/or DAF-12? This would be important to look at to support the notion that hPXR can substitute for the NHRs.
6. In Fig 4H, the various hPXR mutants analyzed are used for NML and RIF treatments. It would be great if the authors could comment on the fact that all mutants have a lower baseline activity. Would this suggest that the compounds stabilize the protein?
7. Fig 5 shows a strong and consistent effect of NML in vivo. While interestingly, this would need to be done in a PXR mutant to suggest that NML uses the detoxification pathway in the context of neuroprotection.

Minor points

1. While the language is clear, I would suggest for a native speaker to edit the manuscript.
2. I would recommend refraining from using the term “anti-aging”. This is not very well defined, particularly in the context of the paper where survival is only measured in the worm. The term better

belongs into popular culture.

3. Also from the data presented here, I think one cannot conclude any recommendations or comments regarding citrus consumption.

Reviewer #2 (Remarks to the Author):

In manuscript "Pregnane X receptor agonist nomilin extends lifespan and healthspan through the detoxification functions", the authors show through a combination of X-ray structure, in vitro and in vivo (in nematode and mouse) experiments that nomilin, a compound found in citrus fruits, is able to activate in a beneficial way for the organism the nuclear receptors NHR8 and DAF12 in worms, as well as their counterpart in mammals, PXR. PXR is a receptor rather known for its adverse effects following the binding of exogenous molecules, whereas here the authors show that nomilin can increase lifespan and healthspan via PXR. From a technical point of view, the biological experiments are numerous and varied, and are rigorously carried out with all necessary controls. The accomplished work seems dense and of quality, even if the authors sometimes get confused in the text, especially for the notations of the mutants. However the quality of the X-ray structure is a major issue. It can be improved and needs to be re-refined. My comments will essentially focus on the structural aspects of this manuscript, even though I have some general comments, listed below, to improve the readability :

1/ Please specify the origin of PXR in the whole manuscript, by writing mPXR (for mouse) or hPXR (for human) depending on the sentence or figure legend.

2/ Fig. 4 : panels D, E, F, and G are not easy to read, please increase the size of residue labels, and potentially the size of these panels. In G, write the name of the compounds with the same color code as the sticks and place the labels in a more obvious way. In H, there is a mistake, M245Q should be M425Q, and M247A should be S247R. In C, please label the b1' strand as it is cited in the text.

Molecules in panel A are too small to be read correctly.

3/ Fig. 6 : panel H, increase the font size.

4/ Please standardize the nomenclature for residues as both « Ser247 and Gln285 » and « M243, W299, I414, and M425 » are found in the manuscript.

5/ Fig. S1 : some panels should be more described, for example add DAY 10 on a and b, DAY 15 on c and d, DAY5 on e and DAY10 on f.

6/ In « Crystal structure of nomilin-PXR complex indicates critical amino acids for the binding affinity » section, in last sentence, M247A/R should be S247A/R.

7/ In « Mammalian PXR is a functional ortholog of NHR-8 /DAF-12 » section there is a mistake on mutant that should be hPXR S247R (I expect)

8/ Maybe I missed something, but in « Discussion », what are AD, PD and CR ?

Here are my scientific comments :

9/ For TR-FRET assay, why the authors used the T0901317 LXR agonist as a reference instead of

rifampicine or SR12813 that are commonly used as PXR reference ligands ? Is this compound known to be a specific PXR agonist ?

In Fig. 4 B, the concentration should be stated as a power of 10 for easier reading.

Could the authors mention and discuss the affinity values (K_i values) of the different compounds in the text ?

Finally, in Fig. S3, could the authors add the equivalent panel to those in a-g for PXR ?

10/ The conclusion of the « Mammalian PXR is a functional ortholog of NHR-8 /DAF-12 » section – « These data indicate that hPXR could partially compensate the function of NHR-8 and DAF-12 in mediating nomilin dependent lifespan-extending effects in *C. elegans* » – implies that hPXR could recognize the response elements of NHR8 and DAF-12. Out of curiosity, are these DNA sequences known in nematode ?

11/ Concerning the structure, it was not clear for me if the authors were able to model the ligand in both chains of PXR structure before reading the PDB validation report and having a look to the structure. Please be more precise in text.

However, several points let me think that this structure should be improved with a new refinement and maybe a second ligand could be added :

- In Table S5 there are some missing classical statistics : $I/\sigma(I)$, redundancy, number of reflections used for refinement, Bfactors and number of atoms for protein, ligands and water. Moreover, the number of unique reflections seems low (5,662) while in the PDB Report the number of reflections in test set is 1,994 (3.27% of total reflections). Please correct this point.

Given statistics and poor quality of some parts of the structure (according to the PDB validation report) let me think that the resolution is overestimated. But without all statistic values (in particular $I/\sigma(I)$) it is difficult to decide (total and high resolution shell).

- A clear panel with an omit map for the ligand is necessary but missing in Fig. 4. It should be added (ligand and density only, not surrounded residues) to clearly understand the conformation of the compound.

- As the structure is already available in PDB, I checked the density in the ligand binding pockets. The ligand in chain A is not fitting the density very well and the difference map indicates some issues. Moreover, the Bfactors are very high and the occupancy of the molecule should be refined. By this way, and by cutting the resolution, the authors would also be able to place the second molecule in chain B. Moreover, the negative difference density (in red) indicates that the acetyl moiety of nomilin is not at the right position, leading to the assumption that the compound in the ligand binding pocket could be the metabolite deacetylnomilin. Could it be possible ?

- Some other issues :

There are too many Ramachandran outliers.

Side chain of Leu308 in chain B could be added.

Lys453 in chain A is not correctly orientated and the big density blob between the two molecules in this region should be modelled with Glu461 B.

Some water molecules could be added.

Met323 A and B are not correctly orientated (maybe alternate positions).

Cys284 B has an alternate position.

A glycerol molecule could be added near Glu282.

His407 seems to have alternate positions.

Etc...

-In sentence « From Fig. 4g, the biphenyl moiety in rifampicin interacts with M243 more closely than nomilin does, making rifampicin more sensitive to the local spatial variation introduced by M425Q mutation », if I understand well, it should be the same Met at the beginning and at the end, no ?

-« These data suggest the nomilin activation on PXR may be different from other known PXR agonist. »

Why is this ? What are the arguments ? Have the authors compared the structure of their complex with all other complex structures of PXR available in PDB ?

Reviewer #3 (Remarks to the Author):

This manuscript by Fan et al. investigates an interesting question – whether a citrus fruit-derived small molecule, nomilin extends healthspan and lifespan in model organisms and the mechanism by which it occurs. The authors use a variety of different assays for this with some exciting results. However, there are some major concerns (listed below).

1. Figure 1: it looks like survival for 50 μ M and 100 μ M NML are comparable. Did the authors test higher doses than these? This would help understand NML-related toxicity (if any).
2. Figure 2: the death rates for each toxicant is very different depending on the mutant. What does this mean? Does NML require either *nhr-8* or *daf-12* for different toxicants? Where does this difference come from? Is it from the mechanism of action of the toxicant?
3. Figure 3: *gst-10* RNAi data shows that NML requires *gst-10* for lifespan extension. Yet, in Figure 2A, *gst10* transcript levels are not significantly elevated by NML. Please explain why this is the case.
4. Figure 4: please identify on the figure that some of these data are related to human cells and some to worms. Presenting everything together can be very confusing. If not, they should be separated/ re-organized.
5. Figure 5: How does the D-galactose induced mouse model relate to the major themes of the manuscript? In other words, were detoxication or longevity genes/ proteins measured in this model in the brains?
6. Figure 6: BDL is not a liver toxicity model, more of a liver inflammation model. Why was not something like APAP or CCl4 not used if the goal was to investigate detoxication? Many of the mechanisms shown in 6H were not directly shown (eg. lifespan extension with NML in mice) therefore this diagram needs to be adjusted to showcase only what was proven with evidence during this study.
7. While the authors are correct in stating that “the increase in detoxication gene expression is a common transcriptomic signature in long-lived worms, flies and rodents”, there is also strong evidence that many of these genes and proteins have higher expression and activity in diseased cells and organs. An example is NQO1 levels in Alzheimer’s Disease brains. Therefore, one has to consider the context and dose under which upregulation of detoxication genes and proteins are beneficial.
8. The doses of nomilin that are needed to “attenuate the efficiency of therapy” in humans may be

much larger than what was tested here. Therefore, it is very difficult to make this case.

Reviewer #4 (Remarks to the Author):

Title: Pregnane X receptor agonist nomilin extends lifespan and healthspan through the detoxification functions

Comments to author:

This paper firstly used nematode *C. elegans* to demonstrate nomilin, a naturally occurring compound in citrus fruits, significantly extended the health span and toxin resistance. Further analysis indicated that the anti-senescence effect of nomilin is dependent on DAF-2/DAF-16 and NHR-30 8/DAF-12. Given hPXR was identified as the mammalian counterpart of NHR-8/DAF-12, the authors further identified that nomilin directly targets hPXR using X-ray crystallography and mutation assays. Finally, the authors demonstrated that nomilin exerts anti-aging effects on D-galactose-induced senescence mice and BDL mice. Overall, this study is interesting and well-organized. However, there are still some concerns that should be clearly addressed.

1. In the Figure 4, the authors used the HEK293T cell-based reporter assay to demonstrate that hPPAR α , hPPAR β , hPPAR γ , FXR, LXR α and NRF2 can't be activated by nomilin. However, other nuclear receptors especially CAR shares high sequence homology with PXR. Whether CAR can be activated by nomilin should be further studied.
2. RNA seq and QPCR analysis revealed that nomilin up-regulated the mRNA levels of hPXR target genes. What the effects of nomilin on the protein expressions of the PXR targets such as CYP3A11, UGT1A1 and SULT2A1?
3. The authors demonstrated that nomilin relieved the liver damage in BDL mice model. A positive control drug should be used to systematically assess the effect of nomilin in BDL-induced liver damage, such as mouse PXR agonist PCN. What's more, the effect of nomilin on liver damage is not sufficient to reflect its role in extending lifespan, other animal models should be used to prove that nomilin extends lifespan and healthspan, for example, senescence related mouse model.
4. There is a mistake in the figure legend of Figure 4, please check and revise carefully.
5. In the Figure 6A, the scale bar is missing and not indicated in the figure legend. There is a mistake in the Figure 6H. The author should check and revise the diagram.

RESPONSE TO REVIEWER COMMENTS

Reviewer #1 (Remarks to the Author):

In this paper, Fan et al investigate the properties of the citrus derived compound nomilin (NML) in nematodes and mice for a protective function in health and ageing. NML robustly extends lifespan and appears to do so through IIS pathway. They then focus on detoxification mechanisms and show that the beneficial effects of NML and nuclear DAF-16 localization depend on *nhr-8* and *daf-12*. In fact, some detoxification genes were required for the NML longevity. Elegantly switching to structural analysis, the authors show that NML binds hPXR and use amino acid substitutions to assess NML interaction in its hPXR binding pocket. Next, they ask if hPXR and NML can affect worm survival. Further analyzing NML in mice, the authors show a neuroprotective effect using two paradigms.

This work shows an impressive breadth of methods and reports very interesting data. Some of the conclusions, however, are not fully supported by the data and will require some additional experimentation.

Essential points:

1. To solidify the effect of NML through IIS and the NHRs, it would be important to check for the expression of known DAF-16 target genes. Does NML affect dauer formation?

We thank the reviewer for the suggestive comments. We examined the mRNA expression of DAF-16 target genes. The results showed that the expression of DAF-16 target genes *sod-2*, *sod-3*, *clk-2*, *acs-19* and *lin-2* in *C. elegans* were increased by nomilin treatment, which support that the effects of nomilin is via IIS signaling. The data were added into Fig. 1f .

We have also performed additional experiment to see whether nomilin may induce dauer formation. We found that nomilin did not enhance the dauer formation either in the WT or in the *daf-2(e1370)* mutant background (Supplementary Figure S1q, r). We

thought that the reason normilin mainly affected longevity instead of dauer formation, possibly because it targeted the intestinal cells and affected the local IIS activity (Fig.1c, d). It was consistent with the report that the intestinal IIS pathway mainly regulates longevity, but not the dauer formation process, while the neuronal IIS pathway does the opposite [Dillin, A., Crawford, D.K., and Kenyon, C. Timing requirements for insulin/IGF-1 signaling in *C. elegans*. Science 298, 830 – 834, 2002]. We have added these data and explanations to the text.

2. For the longevity assay statistics, it would be important to show the number of censored worms. Also, while the authors use large numbers of worms in each experiment, it would be important to do independent biological repeats of the lifespan assays.

We counted the number of censored worms in all lifespan experiments. Now the data were added to Supplementary Table 1-4 and 6. All lifespan experiments were repeated 2-3 time by different investigators. Now the data were shown in Source Data sheet.

3. To solidify the survival experiments, it would be useful to use limonin as a negative control.

We tested the effects of limonin on the lifespan of *C. elegans*, which showed that limonin did not extend the lifespan of *C. elegans* significantly. Now we described it in the text and the data were added as Supplementary Fig. S1b.

4. I see a few inconsistencies in the lifespan data in Fig 4. The lifespan extension in the WT is smaller than that reported in Fig 1. While I really like the idea of testing hPXR in the worm, I do not see the evidence in the data that it can replace NHR-8 or DAF-12. To look at this, there should be a comparison between the “rescue” strains with WT and mutant hPXR in Table S6.

We thank the reviewer's suggestion. Following the reviewer's question, we made several comparisons and found that the mean lifespan of WT PXR and the mutant PXR overexpression animals in the *daf-12(-)* animals under the nomilin supplementation were 18.38 and 16.18 days respectively (p=0.003). While the mean lifespan of WT PXR and the mutant PXR overexpression animals in the *nhr-8(-)*

animals under the nomilin supplementation were 19.96 and 18.63 days respectively ($p=0.07$). These data suggested the ligand binding activity of PXR was statistically significant for mediating the lifespan extension effect of normilin at least in *daf-12(-)*. On the other hand, as the reviewer said, the overexpression of human PXR could not fully restore the lifespan extension effect of nomilin, possibly due to mammalian PXR could not fully activate the *C. elegans* target genes. Now we added these analyses and explanations into the text.

5. What is the effect of hPXR expression on the target genes of NHR-8 and/or DAF-12? This would be important to look at to support the notion that hPXR can substitute for the NHRs.

We performed the experiments to investigate whether hPXR could activate the target genes of NHR-8 and DAF-12. Nomilin-treated hPXR transgenic *nhr-8* or *daf-12* mutants were used to test the mRNA levels. The results showed that nomilin increased the expression of a few genes in these mutants (Fig. 5d, e), suggesting that hPXR partially compensates the function of worm NHR-8 and DAF-12.

6. In Fig 4H, the various hPXR mutants are analyzed are used for NML and RIF treatments. It would be great if the authors could comment on the fact that all mutants have a lower baseline activity. Would this suggest that the compounds stabilize the protein?

We are grateful to the suggestion from the reviewer. These data were obtained in same conditions. Thus, the possible explain for the lower baseline activity of hPXR mutants is that the activities of the mutants in response to the endogenous agonists may be lower than those of wild type hPXR. Currently, we are no evidence to show that the compounds stabilize the protein. Now we added a short comment to the text.

7. Fig 5 shows a strong and consistent effect of NML in vivo. While interestingly, this would need to be done in a PXR mutant to suggest that NML uses the detoxification pathway in the context of neuroprotection.

We fully agree to the comments. We have shown that the effects of nomilin on BDL damage were attenuated in the liver of PXR knockout mice (Fig. 9a). Now we carried out D-galactose induced senescence in PXR knockout mice, and treated with nomilin.

The results showed that nomilin did not improve the cell death in the brain, motor deficits as well as inflammatory infiltration in the liver of PXR knockout mice induced by D-galactose, supporting that NML activates the detoxification pathway in the context of neuroprotection. Now the data were added as Fig. 8.

Minor points

1. While the language is clear, I would suggest for a native speaker to edit the manuscript.

We asked a native English speaker to revise the language.

2. I would recommend refraining from using the term “anti-aging”. This is not very well defined, particularly in the context of the paper where survival is only measured in the worm. The term better belongs into popular culture.

We changed “anti-aging” to “aging inhibiting” or “longevity intervention” through the text.

3. Also from the data presented here, I think one cannot conclude any recommendations or comments regarding citrus consumption.

We revised the relevant conclusion.

Reviewer #2 (Remarks to the Author):

In manuscript "Pregnane X receptor agonist nomilin extends lifespan and healthspan through the detoxification functions", the authors show through a combination of X-ray structure, in vitro and in vivo (in nematode and mouse) experiments that nomilin, a compound found in citrus fruits, is able to activate in a beneficial way for the organism the nuclear receptors NHR8 and DAF12 in worms, as well as their counterpart in mammals, PXR. PXR is a receptor rather known for its adverse effects following the binding of exogenous molecules, whereas here the authors show that nomilin can increase lifespan and healthspan via PXR. From a technical point of view, the biological experiments are numerous and varied, and are rigorously carried out with all necessary controls. The accomplished work seems dense and of quality,

even if the authors sometimes get confused in the text, especially for the notations of the mutants. However the quality of the X-ray structure is a major issue. It can be improved and needs to be re-refined. My comments will essentially focus on the structural aspects of this manuscript, even though I have some general comments, listed below, to improve the readability :

1/ Please specify the origin of PXR in the whole manuscript, by writing mPXR (for mouse) or hPXR (for human) depending on the sentence or figure legend.

We thank the reviewer for the suggestion. Now we specified the origin of PXR through the text.

2/ Fig. 4 : panels D, E, F, and G are not easy to read, please increase the size of residue labels, and potentially the size of these panels. In G, write the name of the compounds with the same color code as the sticks and place the labels in a more obvious way. In H, there is a mistake, M245Q should be M425Q, and M247A should be S247R. In C, please label the b1' strand as it is cited in the text.

Molecules in panel A are too small to be read correctly.

We have re-made the graph and corresponding labels and corrected the mistakes.

3/ Fig. 6 : panel H, increase the font size.

We increased the font size in panel H.

4/ Please standardize the nomenclature for residues as both « Ser247 and Gln285 » and « M243, W299, I414, and M425 » are found in the manuscript.

We standardized the nomenclature for residues.

5/ Fig. S1 : some panels should be more described, for example add DAY 10 on a and b, DAY 15 on c and d, DAY5 on e and DAY10 on f.

We added the information to relevant Figures.

6/ In « Crystal structure of nomilin-PXR complex indicates critical amino acids for the binding affinity » section, in last sentence, M247A/R should be S247A/R.

We corrected the mistakes.

7/ In « Mammalian PXR is a functional ortholog of NHR-8 /DAF-12 » section there is a mistake on mutant that should be hPXR S247R (I expect)

We corrected the mistakes.

8/ Maybe I missed something, but in « Discussion », what are AD, PD and CR ?

We revised “AD, PD and CR” as “Alzheimer's disease, Parkinson's disease and caloric restriction”.

Here are my scientific comments :

9/ For TR-FRET assay, why the authors used the T0901317 LXR agonist as a reference instead of rifampicine or SR12813 that are commonly used as PXR reference ligands ? Is this compound known to be a specific PXR agonist ?

T0901317 is also a high-affinity ligand for PXR, which was suggested by the instruction in the TR-FRET kit. T0901317 binds and activates PXR with the same nanomolar potency with which it stimulates LXR activity (Mitro N, Vargas L, Romeo R, Koder A, Saez E. T0901317 is a potent PXR ligand: implications for the biology ascribed to LXR. FEBS Lett. 2007, 581(9):1721-6.). According to the Instruction in the LanthaScreen® TR-FRET PXR Competitive Binding Assay kit, T0901317 binds to hPXR-LBD with highest affinity similar to SR-121813 (Fig. A), while rifampicin was detected with an IC₅₀ of ~10 μM (Fig. B).

A

B

Figure. Relative IC₅₀ Values of Selected Ligands for PXR-LBD in the LanthaScreen® TR-FRET PXR Competitive Binding Assay. A: T0901317, SR-121813 and other PXR ligands; B: rifampicin.

In Fig. 4 B, the concentration should be stated as a power of 10 for easier reading. The concentration was changed to a power of 10 in Fig. 4b.

Could the authors mention and discuss the affinity values (K_i values) of the different compounds in the text ?

We added the discussion about the affinity values (K_d values) of the compounds to the text.

Finally, in Fig. S3, could the authors add the equivalent panel to those in a-g for PXR ?

PXR transactivity of nomilin was added as Fig. S3a. We also added CAR reporter gene assay as Fig. S3b.

10/ The conclusion of the « Mammalian PXR is a functional ortholog of NHR-8 /DAF-12 » section – « These data indicate that hPXR could partially compensate the function of NHR-8 and DAF-12 in mediating nomilin dependent lifespan-extending effects in *C. elegans* » – implies that hPXR could recognize the response elements of NHR8 and DAF-12. Out of curiosity, are these DNA sequences known in nematode ?

That's a great question. Following reviewer's question, we have checked the published DNA binding site for PXR and their worm homologs. A 2004 JBC paper reported that PXR has multiple consensus DNA binding sequences (including a

relative conserved 3' half site A-G-T-T-C-A sequence) (1). While in a 2004 G&D paper, DAF-12 was also reported to have an similar 3' halfsite A-G-T-T/G-C-A/G DNA binding sequence (2). Though we did not find any report about the binding site of NHR-8, we did find a recent Cell Metabolism paper reporting that “ NHR-8 and DAF-12 share significant homology in DNA- and ligand-binding domains (DBD; LBD), and have identical residues in the P-box, a motif in the first zinc finger that functions in DNA recognition ”(3). Therefore, we think these may explain why hPXR could partially rescue the daf-12(-) mutant (Figure 5).

1. Vyhldal CA, Rogan PK, Leeder JS. Development and refinement of pregnane X receptor (PXR) DNA binding site model using information theory: insights into PXR-mediated gene regulation. *J Biol Chem.* 2004; 279:46779-86.
2. Shostak Y, Gilst MRV, Antebi A, Yamamoto KR. Identification of *C. elegans* DAF-12-binding sites, response elements, and target genes. *Genes Dev.* 2004;18(20):2529-44.
3. Magner DB, Wollam J, Shen Y, Hoppe C, Li D, Latza C, Rottiers V, Hutter H , Antebi A. The NHR-8 nuclear receptor regulates cholesterol and bile acid homeostasis in *C. elegans*. *Cell Metab.* 2013;18(2):212-24.

11/ Concerning the structure, it was not clear for me if the authors were able to model the ligand in both chains of PXR structure before reading the PDB validation report and having a look to the structure. Please be more precise in text.

However, several points let me think that this structure should be improved with a new refinement and maybe a second ligand could be added :

- In Table S5 there are some missing classical statistics : $I/\sigma(I)$, redundancy, number of reflections used for refinement, Bfactors and number of atoms for protein, ligands and water. Moreover, the number of unique reflections seems low (5,662) while in the PDB Report the number of reflections in test set is 1,994 (3.27% of total reflections). Please correct this point.

Given statistics and poor quality of some parts of the structure (according to the PDB validation report) let me think that the resolution is overestimated. But without all statistic values (in particular $I/\sigma(I)$) it is difficult to decide (total and high resolution shell).

We thank the reviewer for the suggestion and re-processed the diffraction data for a

better data reduction and refinement. We discarded several image with poor diffraction and re-scaled the data to a resolution up to 2.1 Å, which is slightly lower than the one in the original manuscript but have the statistics improved significantly. We have renew the crystallographic table with all the statistics required. In addition, we have attached the coordinate file and the reflection file in this revision for further evaluation.

- A clear panel with an omit map for the ligand is necessary but missing in Fig. 4. It should be added (ligand and density only, not surrounded residues) to clearly understand the conformation of the compound.

We have made the omit map for the ligand as figure 4h according to the suggestion.

- As the structure is already available in PDB, I checked the density in the ligand binding pockets. The ligand in chain A is not fitting the density very well and the difference map indicates some issues. Moreover, the Bfactors are very high and the occupancy of the molecule should be refined. By this way, and by cutting the resolution, the authors would also be able to place the second molecule in chain B.

Just as the reply to previous question, we have re-processed the data with a slightly lower resolution for a improved refinement. In the revised model, we successfully fit the ligands in both protomers by occupancy refinement, with a better electron density map in protomer A than that in B (Fig. 4h). We appreciate the reviewer's suggestion and it help us make more accurate model.

Moreover, the negative difference density (in red) indicates that the acetyl moiety of nomilin is not at the right position, leading to the assumption that the compound in the ligand binding pocket could be the metabolite deacetylnomilin. Could it be possible ? We have confirmed the compound structure by a mass spectrometry assay, which showed that the compound is exact nomilin, but not deacetylnomilin (see below Figure 1&2 and Table 1).

Figure 1 The total ion chromatogram of nomilin

Figure 2 The MS/MS fragmentation ion chromatogram of nomilin

Table 1 The identified results of high resolution mass spectrometry of nomilin

Peak	RT(min)	Formula	[M+H] ⁺			MS/MS Fragments	Identified compounds	References
			m/z theory	m/z measured	Error (ppm)			
1	4.23	C ₂₈ H ₃₄ O ₉	515.2276	515.2237	-7.5	469.2191, 369.2020, 187.1747, 161.0579	Nomilin	[1,2]

1. Goh RMV, et al. Investigation of changes in non-traditional indices of maturation in Navel orange peel and juice using GC-MS and LC-QTOF/MS. Food Res Int. 2021;148:110607.
2. Avula B, et al. Liquid Chromatography-Electrospray Ionization Mass Spectrometry Analysis of Limonoids and Flavonoids in Seeds of Grapefruits, Other Citrus Species, and Dietary Supplements. Planta Med. 2016;82(11-12):1058-69.

Now the methods were added to as Supplementary Methods.

- Some other issues :

There are too many Ramachandran outliers.

Side chain of Leu308 in chain B could be added.

Lys453 in chain A is not correctly orientated and the big density blob between the two molecules in this region should be modelled with Glu461 B.

We agreed and have made the modifications accordingly.

Some water molecules could be added.

Met323 A and B are not correctly orientated (maybe alternate positions).

Cys284 B has an alternate position.

A glycerol molecule could be added near Glu282.

His407 seems to have alternate positions.

Etc...

We have corrected the side chain model for M323 and C284. With the new electron density map, the blob of the electron density near E282 is quite small and might not accommodate the glycerol molecule.

-In sentence « From Fig. 4g, the biphenyl moiety in rifampicin interacts with M243 more closely than nomilin does, making rifampicin more sensitive to the local spatial variation introduced by M425Q mutation », if I understand well, it should be the same Met at the beginning and at the end, no ?

Yes, we have corrected the mistake in the residue number.

-« These data suggest the nomilin activation on PXR may be different from other known PXR agonist. » Why is this ? What are the arguments ? Have the authors compared the structure of their complex with all other complex structures of PXR available in PDB ?

We have made a superposition among the hPXR bound with nomilin, rifampicin, SR12813, Hyperforin, and clotrimazole, as well as its empty state (Supplementary Fig. S4a, b). In the superposition between hPXR bound with nomilin and rifampicin, we found the helix of aa 193-209 has a spatial clash with rifampicin. Although this helix can only be visible in our structure, the superposition showed that when bound with rifampicin, there has to be a displacement in this region to avoid clash with the ligand, implying a significant structural difference between hPXR bound with nomilin and rifampicin.

Reviewer #3 (Remarks to the Author):

This manuscript by Fan et al. investigates an interesting question – whether a citrus fruit-derived small molecule, nomilin extends healthspan and lifespan in model organisms and the mechanism by which it occurs. The authors use a variety of different assays for this with some exciting results. However, there are some major concerns (listed below).

1. Figure 1: it looks like survival for 50 μ M and 100 μ M NML are comparable. Did the authors test higher doses than these? This would help understand NML-related toxicity (if any).

We appreciate the reviewer for the comment. We tested 200 μ M of nomilin, which still showed effects on lifespan of *C. elegans*, but mildly decreased, implying a slight side effect. The data were added as Supplementary Fig. S1a.

2. Figure 2: the death rates for each toxicant is very different depending on the mutant. What does this mean? Does NML require either *nhr-8* or *daf-12* for different toxicants? Where does this difference come from? Is it from the mechanism of action of the toxicant?

Those are great questions. We thought the difference could be explained by several reasons. First, the mechanism of toxicity of those toxicants are different. For example, Paraquat enhances Redox pressure, while chloroquine was a lysosome pH neutralizer. Therefore, specific transcriptional activation of relative antioxidant gene were responsible for the related conditions. Second, although NHR-8 and DAF-12 shares similar DNA-binding sequence, they did have independent downstream targets (1), possibly due to the difference of their expression pattern, and their transcription activation preference of targeted genes. Third, it is well known that many ligands of nuclear receptors can often regulate different genes in different cells/tissues based on the availability of coregulators (2,3).

1. Magner DB, Wollam J, Shen Y, et al. The NHR-8 nuclear receptor regulates cholesterol and bile acid homeostasis in *C. elegans*. *Cell Metab.* 2013;18(2):212-24.

2. Frigo DE, Bondesson M, Williams C. Nuclear receptors: from molecular mechanisms to therapeutics. *Essays Biochem.* 2021; 65 (6): 847–856.

3. Martinkovich S, Shah D, Planey SL and Arnott JA. Selective estrogen receptor modulators: tissue specificity and clinical utility. *Clin. Interv. Aging* 2014; 9, 1437–1452.

3. Figure 3: *gst-10* RNAi data shows that NML requires *gst-10* for lifespan extension. Yet, in Figure 2A, *gst10* transcript levels are not significantly elevated by NML. Please explain why this is the case.

Gst-10 is a typo. We are sorry for the mistake. We selected to knockdown the nomilin upregulated gene *Gst-4* (Fig. 2a) to verify the pathway. Now we corrected the mistake.

4. Figure 4: please identify on the figure that some of these data are related to human cells and some to worms. Presenting everything together can be very confusing. If not, they should be separated/ re-organized.

We re-organized the Figure as Figure 4 and 5 to separate human and *C. elegans* data.

5. Figure 5: How does the D-galactose induced mouse model relate to the major themes of the manuscript? In other words, were detoxication or longevity genes/ proteins measured in this model in the brains?

We thank the reviewer for the helpful suggestions. The liver is a major detoxification organ in mammals, which may detoxify D-galactose, and lower the concentration of D-galactose in circulation and the organs. However, PXR is also expressed in the brain, which may play detoxification functions. We analyzed PXR target gene expression and showed that nomilin increased the expression of *Gsta1*, *Gsta2*, *Mdr3*, *Cyp8b1*, *Cyp27a1* and *Cyp2d22* in the hippocampus, suggesting that nomilin may also increase PXR signaling in the brain of mice. Now the data were added as Supplementary Fig. S6d.

6. Figure 6: BDL is not a liver toxicity model, more of a liver inflammation model. Why was not something like APAP or CCl4 not used if the goal was to investigate detoxication? Many of the mechanisms shown in 6H were not directly shown (eg.

lifespan extension with NML in mice) therefore this diagram needs to be adjusted to showcase only what was proven with evidence during this study.

BDL may cause cholestasis manifested oxidative stress response, extensive hepatocyte necrosis and inflammation, which results from intrahepatic retention of toxic hydrophobic bile salts (1,2). Many studies have identified that PXR may protect cholestatic hepatotoxicity through detoxifying toxic hydrophobic bile acids (3). Cholestatic PXR knockout mice exhibits more hepatic damage than wild-type mice both after bile duct ligation and cholic acid feeding [4-6]. PXR agonist reduces lithocholic acid-induced liver injury in wild-type mice, but not in PXR knockout mice, via the upregulation of UGT1A1, MRP2, MRP3 and CYP3A4 facilitating bilirubin elimination and detoxification of bile acids [6,7]. The summary diagram has been revised accordingly.

1. Beuers U, Boyer JL, & Paumgartner G. Ursodeoxycholic acid in cholestasis: potential mechanisms of action and therapeutic applications. *Hepatology*, 1998,28, 1449-1453.
2. Woolbright BL, Antoine DJ, Jenkins RE, Bajt ML, Park BK, Jaeschke H. Plasma biomarkers of liver injury and inflammation demonstrate a lack of apoptosis during obstructive cholestasis in mice. *Toxicology and Applied Pharmacology*, 2013,273, 524-531.
3. Kakizaki S, Takizawa D, Tojima H, Yamazaki Y, Mori M. Xenobiotic-sensing nuclear receptors CAR and PXR as drug targets in cholestatic liver disease. *Curr Drug Targets*. 2009,10:1156-1163.
4. Stedman CA, Liddle C, Coulter SA, Sonoda J, Alvarez JG, Evans RM, et al. Benefit of farnesoid X receptor inhibition in obstructive cholestasis. *Proc Natl Acad Sci U S A*, 2006, 103:11323-11328.
5. Stedman CA, Liddle C, Coulter SA, Sonoda J, Alvarez JG, Moore DD, et al. Nuclear receptors constitutive androstane receptor and pregnane X receptor ameliorate cholestatic liver injury. *Proc Natl Acad Sci U S A*, 2005, 102:2063-2068.
6. Teng S, Piquette-Miller M. Hepatoprotective role of PXR activation and MRP3 in cholic acid-induced cholestasis. *Br J Pharmacol*, 2007, 151:367-376.
7. Marschall HU, Wagner M, Zollner G, Fickert P, Diczfalusy U, Gumhold J, et al. Complementary stimulation of hepatobiliary transport and detoxification systems by rifampicin and ursodeoxycholic acid in humans. *Gastroenterology*, 2005, 129:476-485.

7. While the authors are correct in stating that “the increase in detoxication gene expression is a common transcriptomic signature in long-lived worms, flies and rodents”, there is also strong evidence that many of these genes and proteins have higher expression and activity in diseased cells and organs. An example is NQO1 levels in Alzheimer’s Disease brains. Therefore, one has to consider the context and dose under which upregulation of detoxication genes and proteins are beneficial.

We really appreciate the reviewer's suggestion. As the reviewer said, in many biological processes, the specific context of application and related gene/protein dosage is critical for their beneficial function. For example, the elevation of NQO1 is associated with AD pathology, but it is commonly viewed as a neuroprotective response to the oxidative stress that accompanies AD (1,2). Therefore, a causative analysis is needed to identify the positive/negative role of an elevation of gene/protein in a specific biological process. That's why in this manuscript, we have shown that NHR-8/DAF-12/PXR activation was not only related, but also required for the lifespan extension and detoxification process in *C. elegans* and mice.

1. Ross D, Siegel D. The diverse functionality of NQO1 and its roles in redox control. *Redox Biol.* 202;41:101950.
2. Lee W-S, Ham W, Kim J. Roles of NAD(P)H:quinone Oxidoreductase 1 in Diverse Diseases. *Life.* 2021;11(12):1301.

8. The doses of nomilin that are needed to “attenuate the efficiency of therapy” in humans may be much larger than what was tested here. Therefore, it is very difficult to make this case.

We thank the reviewer for pointing this out. Although the citrus juices and grapefruit juices contain high concentrations of limonoids (320 ppm and 195 ppm), the effects of nomilin on drug metabolism remains unclear. Now we are working on this, hopefully would answer the question. We revised the sentence to “Whether the consumption of nomilin-containing citrus fruits and juices change drug metabolism needs to be investigated”.

Reviewer #4 (Remarks to the Author):

Title: Pregnane X receptor agonist nomilin extends lifespan and healthspan through the detoxification functions

Comments to author:

This paper firstly used nematode *C. elegans* to demonstrate nomilin, a naturally occurring compound in citrus fruits, significantly extended the health span and toxin resistance. Further analysis indicated that the anti-senescence effect of nomilin is dependent on DAF-2/DAF-16 and NHR-30 8/DAF-12. Given hPXR was identified as the mammalian counterpart of NHR-8/DAF-12, the authors further identified that nomilin directly targets hPXR using X-ray crystallography and mutation assays. Finally, the authors demonstrated that nomilin exerts anti-aging effects on D-galactose-induced senescence mice and BDL mice. Overall, this study is interesting and well-organized. However, there are still some concerns that should be clearly addressed.

1. In the Figure 4, the authors used the HEK293T cell-based reporter assay to demonstrate that hPPAR α , hPPAR β , hPPAR γ , FXR, LXR α and NRF2 can't be activated by nomilin. However, other nuclear receptors especially CAR shares high sequence homology with PXR. Whether CAR can be activated by nomilin should be further studied.

We thank the reviewer for pointing this out. We created pcDNA3.1-hCAR expression plasmid and performed reporter gene assay. The results showed that nomilin did not activate hCAR transactivity. The data was added as Supplementary Fig. S3b.

2. RNA seq and QPCR analysis revealed that nomilin up-regulated the mRNA levels of hPXR target genes. What the effects of nomilin on the protein expressions of the PXR targets such as CYP3A11, UGT1A1 and SULT2A1?

We agree with the comments from the reviewer. We carried out western blots of PXR targets Gsta1, Cyp3a11 and Cyp51a1 in the livers in D-gal-treated mice, and the

results showed that nomilin also increased the protein expression, which consistent to the RT-PCR results. The data were added as Supplementary Fig. S5b, c.

3. The authors demonstrated that nomilin relieved the liver damage in BDL mice model. A positive control drug should be used to systematically assess the effect of nomilin in BDL-induced liver damage, such as mouse PXR agonist PCN. What's more, the effect of nomilin on liver damage is not sufficient to reflect its role in extending lifespan, other animal models should be used to prove that nomilin extends lifespan and healthspan, for example, senescence related mouse model.

PCN counteracts cholestasis by reducing serum levels of bilirubin and bile acids, and inducing bile acid-hydroxylating/detoxifying enzymes Cyp3a11 and Cyp2b10, bile acid transporters Mrp3 and Oatp2 in BDL mice (1). Also, PCN has been reported to reduce lithocholic acid-induced liver injury in mice via the upregulation of the basolateral BA efflux transporter MRP3 (2,3), suggesting that PCN may stimulate major hepatic bile acid/bilirubin metabolizing and detoxifying enzymes and hepatic key alternative efflux systems. We performed mouse BDL experiments and treated with PCN as a positive control. The results showed that PCN improved the necrosis in the liver of mice, which similar to the previous reports. And nomilin showed similar hepatic protective effects as seen in PCN-treated mice. Now the data were added to Fig. 9a.

1. Wagner M, Halilbasic E, Marschall HU, Zollner G, Fickert P, Langner C, Zatloukal K, Denk H, Trauner M. CAR and PXR agonists stimulate hepatic bile acid and bilirubin detoxification and elimination pathways in mice. *Hepatology*. 2005;42(2):420-30.

2. Staudinger JL, Goodwin B, Jones SA, Hawkins-Brown D, MacKenzie KI, LaTour A, et al. The nuclear receptor PXR is a lithocholic acid sensor that protects against liver toxicity. *Proc Natl Acad Sci USA*, 2001;98:3369-3374

3. Teng S, Piquette-Miller M. Hepatoprotective role of PXR activation and MRP3 in cholic acid-induced cholestasis. *Br J Pharmacol*, 2007, 151:367-376.

The chemotherapeutic drug doxorubicin may induce accelerated aging in patients, and has been used to induce senescent animal models in aging research. Thus, we used doxorubicin-induced accelerating aging mouse model to test the effects of nomilin on

lifespan and healthspan. The results showed that nomilin could improve the lifespan and movement ability, liver damage and heart fibrosis in doxorubicin-treated mice, which further support that nomilin exerts aging-retarding effects via the detoxification function. The data were shown as Fig. 7.

4. There is a mistake in the figure legend of Figure 4, please check and revise carefully.

We revised the mistakes.

5. In the Figure 6A, the scale bar is missing and not indicated in the figure legend. There is a mistake in the Figure 6H. The author should check and revise the diagram.

We added the scale bar to the Fig. now as Fig. 9a, and the diagram was revised.

REVIEWER COMMENTS

Reviewer #1 (Remarks to the Author):

All points have been addressed and the paper was enhanced by significant new data. I am in full support of publication.

Reviewer #2 (Remarks to the Author):

I thank the authors for implementing the majority of the corrections I suggested and for their effort to re-process the structural data. The quality of the structure has been improved. However I noticed remaining major concerns.

1/ The alternate position of His407, which is clearly indicated by the difference electron density, is still missing. I insist on that point because of the location of this residue, near the ligand, which is known to adopt several orientations depending of the ligand bound to PXR. Moreover the authors modeled one orientation in one chain, and the other one in the other chain of the asymmetric unit.

2/ The authors added an argument supported by Supp Fig. S4a to discuss the difference of activation levels observed for nomilin and rifampicine respectively (line 246 p7). "In addition, one of the major structural differences between hPXR bound with nomilin and rifampicin is the helix formed by amino acids 193–209, which is well-refined in our structure but absent in a previous report (Supplementary Fig. S4a) 59-62." References 59-62 refer to publications linked to structures used for the superposition seen in Supp Fig. S4a where authors affirm that helix 193-209 is visible only in their structure and not in all other five for which they intentionnally added dashed-lines to symbolize the missing helix: "The helix comprising with amino acid 193-209 was highlighted in dark blue in hPXR-nomilin stucture and the corresponding parts unsolved in other structures were indicated with dot lines." This is totally wrong, this helix is only missing in the structure with rifampicine. I don't understand the goal of authors at this point?!

I also noticed some other minor points.

3/ Concerning the TR-FRET assay, the authors added Kd values for the tested compounds on the graphs in Fig. 4b. Where do they come from? Did the authors measure them by a direct method? In the Mat&Meth section authors say: "The IC50 was calculated using GraphPad Prism 8 and Ki was fitted according to the equation: $K_i = IC_{50}/(1 + [tracer]/K_D)$ ». I missed something...

4/ The discussion below concerning the response elements could be added into the manuscript to support the conclusion of section 'Mammalian PXR is a functional ortholog of NHR-8 /DAF-12'

That's a great question. Following reviewer's question, we have checked the published DNA binding site for PXR and their worm homologs. A 2004 JBC paper reported that PXR has multiple consensus DNA binding sequences (including a relative conserved 3' half site A-G-T-T-C-A sequence) (1). While in a 2004 G&D paper, DAF-12 was also reported to have an similar 3' halfsite A-G-T-T/G-C-A/G DNA binding sequence (2). Though we did not find any report about the binding site of NHR-8, we did find a recent Cell Metabolism paper reporting that " NHR-8 and DAF-12 share significant homology in DNA- and ligand-binding domains (DBD; LBD), and have identical residues in the P-box, a motif in the first zinc finger that functions in DNA recognition "(3). Therefore, we think these may explain why hPXR could partially rescue the daf-12(-) mutant (Figure 5).

1. Vyhldal CA, Rogan PK, Leeder JS. Development and refinement of pregnane X receptor (PXR) DNA binding site model using information theory: insights into PXR-mediated gene regulation. J Biol Chem. 2004; 279:46779-86.
2. Shostak Y, Gilst MRV, Antebi A, Yamamoto KR. Identification of C. elegans DAF-12-binding sites, response elements, and target genes. Genes Dev. 2004;18(20):2529-44.
3. Magner DB, Wollam J, Shen Y, Hoppe C, Li D, Latza C, Rottiers V, Hutter H , Antebi A. The NHR-8 nuclear receptor regulates cholesterol and bile acid homeostasis in C. elegans. Cell Metab. 2013;18(2):212-24.

5/ Just a remark, the mass spectrometry control should have been done directly on PXR/nomilin crystals as the compound could have been modified during the crystallization process. It happens some times.

"We have confirmed the compound structure by a mass spectrometry assay, which showed that the compound is exact nomilin, but not deacetylnomilin (see below Figure 1&2 and Table 1)."

6/ Line 237 p7, G285 should be Q285.

7/ Fig4. i, the mutation of W299 is missing.

Reviewer #3 (Remarks to the Author):

Here are my comments to the author responses -

1. Please increase the quality of the supplemental figures so that the details of figures can be clearly seen.
2. OK
3. OK
4. OK
5. I would switch Figure 7 and 8 - since 6 is about D-galactose
6. OK
7. Lifespan extension in mice was only shown in a heavy toxicity model. Related to this, in Figure 7A.

Please provide a Kaplan-Meier survival curve and corresponding statistical analysis (Long-rank test).

Please provide details on the following related to the Dox model:

- 7H. Please label the trichrome figures and also provide figures that show comparable regions for all groups

- Please provide gait parameters for these mice as they are shown in the other models

- Please provide any body weight and food consumption data for this model as well as other mouse models.

8. OK

Reviewer #4 (Remarks to the Author):

Overall, the authors have made a significant effort to address the reviewer's concern and thereby substantially improved the manuscript. However, some points still need clarification as they have not been corroborated sufficiently.

Major points:

1. One major concern is that the authors should explain why they choose doxorubicin-induced accelerating aging mice as the senescence model. In the main text, the authors proposed "This drug has been used to induce senescence in an animal model in aging research [72]". However, the cited reference is "An evaluation of hepatotoxicity in breast cancer patients receiving injection Doxorubicin", which is about the hepatotoxicity and doxorubicin. Why did the authors not use more classical senescence mouse models, such as SAMP or naturally aging mice. Additionally, in the Figure 7, nomilin promoted the survival of doxorubicin-treated mice and suppressed inflammatory cell infiltration in the liver. However, these data only suggested that nomilin ameliorated the liver injury induced by doxorubicin. At least, the author should perform the analysis of senescence-associated secretory phenotype, for example, p16ink4a, IL1 α , and IL6.

2. In Fig.6, the authors used D-gal to induce senescence. However, there is still a lack of behavioral experiments to reflect the hippocampus and neuromuscular functions.

Response to the reviewer's comments

Reviewer #1 (Remarks to the Author):

All points have been addressed and the paper was enhanced by significant new data. I am in full support of publication.

We thank the reviewer for the positive comments.

Reviewer #2 (Remarks to the Author):

I thank the authors for implementing the majority of the corrections I suggested and for their effort to re-process the structural data. The quality of the structure has been improved. However I noticed remaining major concerns.

1/ The alternate position of His407, which is clearly indicated by the difference electron density, is still missing. I insist on that point because of the location of this residue, near the ligand, which is known to adopt several orientations depending of the ligand bound to PXR. Moreover the authors modeled one orientation in one chain, and the other one in the other chain of the asymmetric unit.

We thank the reviewer for the suggestion, and we have re-processed the data to add the alternative conformation of H407, in both protomers. We have modified Figure 4D with the alternative conformations, and the RCSB deposition has been updated accordingly.

2/ The authors added an argument supported by Supp Fig. S4a to discuss the difference of activation levels observed for nomilin and rifampicine respectively (line 246 p7). “In addition, one of the major structural differences between hPXR bound with nomilin and rifampicin is the helix formed by amino acids 193–209, which is well-refined in our structure but absent in a previous report (Supplementary Fig. S4a) 59-62.” References 59-62 refer to publications linked to structures used for the superposition seen in Supp Fig. S4a where authors affirm that helix 193-209 is visible only in their structure and not in all other five for which they intentionnally added dashed-lines to symbolize the missing helix: “The helix comprising with amino acid 193-209 was highlighted in dark blue in hPXR-nomilin stucture and the corresponding parts unsolved in other structures were indicated

with dot lines.” This is totally wrong, this helix is only missing in the structure with rifampicine. I don’t understand the goal of authors at this point?!

We thank the reviewer for pointing out the incorrect description in the main text and the mistake in Figure S4a. We have revised the paragraph to accurately reflect the correct information and have updated Figure S4a accordingly. We apologize for any confusion caused by the previous incorrect description. We are grateful for the reviewer's input, which has helped improve the accuracy and integrity of our study.

I also noticed some other minor points.

3/ Concerning the TR-FRET assay, the authors added K_d values for the tested compounds on the graphs in Fig. 4b. Where do they come from? Did the authors measure them by a direct method? In the Mat&Meth section authors say: “The IC₅₀ was calculated using GraphPad Prism 8 and K_i was fitted according to the equation: $K_i = IC_{50}/(1 + [tracer]/K_D)$ ». I missed something...

We thank the reviewer for pointing this issues out. Now we have described it in detail in Methods: TR-FRET assay. The dissociation constant (K_d) was fitted into a one-site total binding saturation equation in GraphPad Prism software according to the previous described method (1). IC₅₀ values were determined using log (inhibitor) vs. response - Variable slope model fit by GraphPad Prism software.

1. Lin W, Chen T. Using TR-FRET to investigate protein-protein interactions. **Adv Protein Chem Struct Biol.** 2018;110:31-63.

4/ The discussion below concerning the response elements could be added into the manuscript to support the conclusion of section ‘Mammalian PXR is a functional ortholog of NHR-8 /DAF-12’

That’s a great question. Following reviewer’s question, we have checked the published DNA binding site for PXR and their worm homologs. A 2004 JBC paper reported that PXR has multiple consensus DNA binding sequences (including a relative conserved 3’ half site A-G-T-T-C-A sequence) (1). While in a 2004 G&D paper, DAF-12 was also reported to have an similar 3’ halfsite A-G-T-T/G-C-A/G DNA binding sequence (2). Though we did not find

any report about the binding site of NHR-8, we did find a recent Cell Metabolism paper reporting that “NHR-8 and DAF-12 share significant homology in DNA- and ligand-binding domains (DBD; LBD), and have identical residues in the P-box, a motif in the first zinc finger that functions in DNA recognition ”(3). Therefore, we think these may explain why hPXR could partially rescue the daf-12(-) mutant (Figure 5).

1. Vyhlidal CA, Rogan PK, Leeder JS. Development and refinement of pregnane X receptor (PXR) DNA binding site model using information theory: insights into PXR-mediated gene regulation. *J Biol Chem.* 2004; 279:46779-86.
2. Shostak Y, Gilst MRV, Antebi A, Yamamoto KR. Identification of *C. elegans* DAF-12-binding sites, response elements, and target genes. *Genes Dev.* 2004;18(20):2529-44.
3. Magner DB, Wollam J, Shen Y, Hoppe C, Li D, Latza C, Rottiers V, Hutter H , Antebi A. The NHR-8 nuclear receptor regulates cholesterol and bile acid homeostasis in *C. elegans*. *Cell Metab.* 2013;18(2):212-24.

We thank the reviewer for the helpful suggestions. Now we added it to the manuscript on Page 8.

5/ Just a remark, the mass spectrometry control should have been done directly on PXR/nomilin crystals as the compound could have been modified during the crystallization process. It happens some times.

"We have confirmed the compound structure by a mass spectrometry assay, which showed that the compound is exact nomilin, but not deacetylnomilin (see below Figure 1&2 and Table 1)."

We appreciate the reviewer's comment regarding the potential chemical modification of nomilin during the crystallization process. As per the recommendation, we have remade the crystallization trial and conducted mass spectrometry analysis directly to ensure the integrity of the compound during the crystallization process. Our results confirm that there was no significant deacetylation detected, validating the findings reported in our previous revision. The data were shown as Supplementary Methods-Figure 1a-d and Table 1. Thank you for bringing this to our attention and allowing us to further verify the robustness of our experimental approach.

6/ Line 237 p7, G285 should be Q285.

We thank the reviewer for pointing out this typo, and it has been corrected in the revised manuscript.

7/ Fig4. i, the mutation of W299 is missing.

We have changed W299 to W299R in Fig. 4i.

Reviewer #3 (Remarks to the Author):

Here are my comments to the author responses -

1. Please increase the quality of the supplemental figures so that the details of figures can be clearly seen.

We have prepared high resolution figures.

2. OK

3. OK

4. OK

5. I would switch Figure 7 and 8 - since 6 is about D-galactose.

We are grateful to the suggestions. Now we have switch Fig. 7 and 8.

6. OK

7. Lifespan extension in mice was only shown in a heavy toxicity model. Related to this, in Figure 7A. Please provide a Kaplan-Meier survival curve and corresponding statistical analysis (Long-rank test). Please provide details on the following related to the Dox model:

- 7H. Please label the trichrome figures and also provide figures that show comparable regions for all groups

- Please provide gait parameters for these mice as they are shown in the other models

- Please provide any body weight and food consumption data for this model as well as other mouse models.

We thank the reviewer for the comments and suggestions. Now we have added the Kaplan-Meier survival curve and corresponding statistical analysis (Long-rank test) as Supplementary Fig. S7.

We labeled the trichrome figures and provided figures that show comparable regions for all groups (Fig. 8h).

We provided additional gait parameters in PXR^{-/-} mice (Fig. 7e-f). However, after two-weeks treatment, doxorubicin mice showed significant body weight reduction and poor physical condition that prevented them from completing subsequent behavioral experiment, so we could not obtain the gait parameters in this model.

We provided body weight data for all four mouse models. We also provided the food consumption data of three models (Supplementary Fig. S9). However, for doxorubicin-treated mice, the food was soaked in water to facilitate their eating since the mice were very weak. Thus, the food intake cannot be counted accurately in this model.

8. OK

Reviewer #4 (Remarks to the Author):

Overall, the authors have made a significant effort to address the reviewer's concern and thereby substantially improved the manuscript. However, some points still need clarification as they have not been corroborated sufficiently.

Major points:

1. One major concern is that the authors should explain why they choose doxorubicin-induced accelerating aging mice as the senescence model. In the main text, the authors proposed “This drug has been used to induce senescence in an animal model in aging research [72]”. However, the cited reference is “An evaluation of hepatotoxicity in breast cancer patients receiving injection Doxorubicin”, which is about the hepatotoxicity and doxorubicin. Why did the authors not use more classical senescence mouse models, such as SAMP or naturally aging mice.

Additionally, in the Figure 7, nomilin promoted the survival of doxorubicin-treated mice and suppressed inflammatory cell infiltration in the liver. However, these data only suggested that nomilin ameliorated the liver injury induced by doxorubicin. At least, the author should perform the analysis of senescence-associated secretory phenotype, for example, p16^{ink4a}, IL1 α , and IL6.

We thank the reviewer for the comments and suggestions. Doxorubicin can induce cellular and organ senescence in many aspects as well as the biomarkers of senescence such as p16^{INK4A}, p21 and β -galactosidase in rodents and humans (1, 2). This model was widely

used in aging research. We are sorry for inappropriate reference citing. Now we have cited relevant references in the text.

1. Baar MP, Brandt RMC, Putavet DA, Klein JDD, Derks KWJ, Bourgeois BRM, Stryeck S, Rijksen Y, van Willigenburg H, Feijtel DA, van der Pluijm I, Essers J, van Cappellen WA, van IJcken WF, Houtsmuller AB, Pothof J, de Bruin RWF, Madl T, Hoeijmakers JHJ, Campisi J, de Keizer PLJ. Targeted apoptosis of senescent cells restores tissue homeostasis in response to chemotoxicity and aging. *Cell*. 2017,169(1):132-147.
2. Sun T, Zhang L, Feng J, Bao L, Wang J, Song Z, Mao Z, Li J, Hu Z. Characterization of cellular senescence in doxorubicin-induced aging mice. *Exp Gerontol*. 2022, 163:111800.

We have performed additional animal experiments using SAMP8 accelerated senescence mice. Our results showed that nomilin could ameliorate motor impairment and anxiety-like behavior, and increase learning and memory abilities in SAMP8 mice, indicating that nomilin could extend healthspan in SAMP8 mice. Now the contents were added to the text as section “Nomilin extends healthspan in SAMP8 mice”, and the data were shown as Fig. 9a-m. During this experiment, the survival curves of the two groups of mice are still being observed. At 37-weeks old, 4/9 of the control SAMP8 mice died, while only 2/10 of nomilin-treated SAMP8 mice died. If lifespan would show significant changes, we will add the data to the next version of the text.

We performed the expression analysis of senescence-associated secretory phenotype genes, including *p16^{INK4A}*, *Nlrp3*, *Tnfa*, *Il1a*, *Mcp1* and *Il6*, in the liver of doxorubicin-induced senescent mice. The results showed that doxorubicin-induced the expression of *Nlrp3*, *Tnfa*, *Il-6*, *Il-1β*, *Mcp1* and *p16^{INK4A}*, while nomilin attenuated the expression of *Il6* and *Nlrp3*. Now the data were cooperated to the text as shown as Fig. 8f. In addition, we determined the expression levels of PXR downstream genes in the liver of doxorubicin-induced mice. The results showed that the mRNA levels of *Cyp3a11*, *Por*, *Gsta1/2* and *Mdr3* were increased by nomilin intervention. Now the data were added as Fig. 8g.

2. In Fig.6, the authors used D-gal to induce senescence. However, there is still a lack of behavioral experiments to reflect the hippocampus and neuromuscular functions. We thank the reviewer for the comments and suggestions. We carried out short-/long-term memory tests that correlated with the change of the hippocampus during aging. The exploration

time (Fig. 6c, e) and performance rate (Fig. 6d, f) of mice were significantly improved by nomilin treatment in both short-/long-term memory tests. Now the data were added as Fig. 6c-f. In addition, the pole test, the beam balance test and gait analysis are all methods for assay the movement disorders (1-3), which may reflect neuromuscular functions in the mice (Fig. 6g-n).

1. Ziegler CG, Peng M, Falk MJ, Polyak E, Tsika E, Ischiropoulos H, Bakalar D, Blendy JA, Gasser DL. Parkinson's disease-like neuromuscular defects occur in prenyl diphosphate synthase subunit 2 (Pdss2) mutant mice. **Mitochondrion**. 2012 Mar;12(2):248-57.
2. Häkkinen A, Holopainen E, Kautiainen H, Sillanpää E, Häkkinen K. Balance balance Neuromuscular function and balance of prepubertal and pubertal blind and sighted boys. **Acta Paediatr**. 2006, 95(10):1277-83.
3. Kang DW, Choi JG, Moon JY, Kang SY, Ryu Y, Park JB, Kim HW. Automated Gait Analysis in Mice with Chronic Constriction Injury. **J Vis Exp**. 2017,(128):56402.

REVIEWERS' COMMENTS

Reviewer #1 (Remarks to the Author):

I have assessed the authors' responses to the additional points raised by Reviewer 3. The authors have addressed those points fully.

Reviewer #2 (Remarks to the Author):

All my comments were taken into account and the structure and its interpretation were considerably improved. I support the publication of this work which highlights for the first time the positive effects that PXR could have.

Reviewer #4 (Remarks to the Author):

All points have been addressed and I agree to the publication.